# MIRA: Memory-Integrated Reinforcement Learning Agent with Limited LLM Guidance

**Narjes Nourzad**
University of Southern California
Electrical and Computer Engineering
Los Angeles, CA 90007, USA
`nourzad@usc.edu`

**Carlee Joe-Wong**
Carnegie Mellon University
Electrical and Computer Engineering
Pittsburgh, PA 15213, USA
`cjoewong@andrew.cmu.edu`

## Abstract

Reinforcement learning (RL) agents often face high sample complexity in sparse or delayed reward settings, due to limited prior knowledge. Conversely, large language models (LLMs) can provide subgoal structures, plausible trajectories, and abstract priors that support early learning. Yet heavy reliance on LLMs introduces scalability issues and risks dependence on unreliable signals, motivating ongoing efforts to integrate LLM guidance without compromising RL's autonomy. We propose MIRA (Memory-Integrated Reinforcement Learning Agent), which incorporates a structured, evolving *memory graph* to guide early learning. This graph stores decision-relevant information, such as trajectory segments and subgoal decompositions, and is co-constructed from the agent's high-return experiences and LLM outputs, amortizing LLM queries into a persistent memory instead of relying on continuous real-time supervision. From this structure, we derive a *utility* signal that softly adjusts advantage estimation to refine policy updates without altering the underlying reward function. As training progresses, the agent's policy surpasses the initial LLM-derived priors, and the utility term decays, leaving long-term convergence guarantees intact. We show theoretically that this utility-based shaping improves early-stage learning in sparse-reward settings. Empirically, MIRA outperforms RL baselines and reaches returns comparable to methods that rely on frequent LLM supervision, while requiring substantially fewer online LLM queries[1].

## 1 Introduction

Reinforcement learning (RL) models sequential decision-making as interactions with an environment and learns behavior from reward-driven feedback. RL has achieved strong results in domains including robotic manipulation, dynamic scheduling, and autonomous planning (Nourzad et al., 2024; Liu et al., 2024; Luo et al., 2024). However, these advances often rely on environments with dense, readily accessible rewards. In many tasks, rewards are sparse or delayed, appearing only when specific goals are reached or several steps after the action unfolds. These weak or infrequent rewards obscure which past actions influenced the outcome, making it difficult to credit the eventual reward appropriately (Velu et al., 2023). This uncertainty weakens the gradient signal, leaving policy updates underinformed. Thus, agents become highly data-hungry and require large numbers of interactions to learn useful behaviors (Devidze et al., 2022). These challenges intensify under partial observability, as agents must generalize from limited state information and often struggle early in training (Hausknecht & Stone, 2015; Kurniawati, 2022). In such settings, random exploration rarely uncovers informative trajectories, leading to slow convergence and high variance in returns.

Large language models (LLMs) provide a complementary source of prior knowledge, especially in environments where rewards are sparse, feedback is delayed, and observations are partial. They have demonstrated capabilities in reasoning over abstract goals, interpreting high-level intent, and leveraging broad prior knowledge (Jimenez et al., 2023; Xu et al., 2024). These properties make them

---

[1]Project webpage : `https://narjesno.github.io/MIRA/`

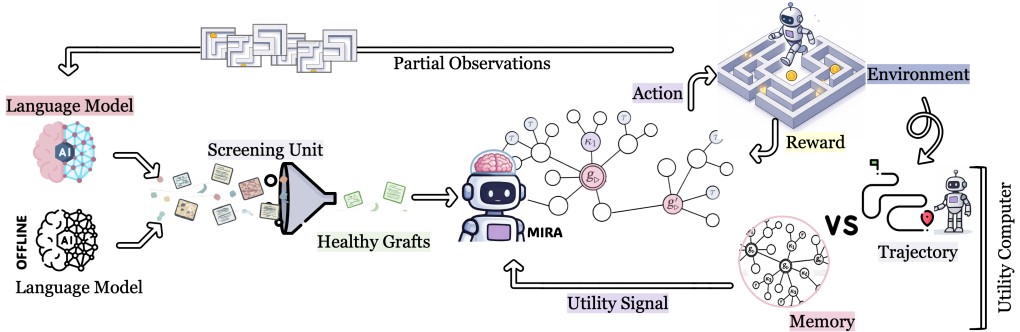

Figure 1: Offline priors and online LLM suggestions are filtered by a screening unit before being incorporated into the memory graph as *healthy grafts*. MIRA agent acts under partial observations, interacting with the environment. A utility module evaluates trajectory rollouts against the evolving memory graph, producing a utility signal that shapes advantage estimation and policy updates.

natural candidates for providing structured guidance for RL agents (Schoepp et al., 2025). A growing body of work has explored how pretrained LLMs can support RL to improve sample efficiency. One line of research positions the LLM as an implicit or explicit reward model, either estimating reward signals from environment descriptions or generating code to define reward functions (Ma et al., 2025; Kwon et al., 2023; Fan et al., 2022; Rocamonde et al., 2023; Bhambri et al., 2024; Xie et al., 2024). Another line leverages LLMs to generate high-level plans, policy sketches, or step-by-step guidance during training (Du et al., 2023; Hu & Sadigh, 2023; Dasgupta et al., 2023; Wang et al., 2023; Zhou et al., 2023). A third direction focuses on task-level guidance such as subgoal decomposition, curriculum design, or goal interpretation from natural language (Wang et al., 2024a; Ma et al., 2023; Shinn et al., 2023). Additional related approaches appear in Appendix B.

RESEARCH CHALLENGES. The existing approaches, although promising, typically require *frequent (often per-step) LLM supervision*, making the agent's performance heavily reliant on LLM inference, which introduces several difficulties. First, it can interfere with the RL learning signal (Zhou et al., 2023), limiting autonomous decision-making and reducing the agent's ability to generalize or adapt if the LLM becomes unavailable. Second, since LLMs cannot interact directly with the environment or gather real-time feedback, full reliance on their instructions is suboptimal (Qu et al., 2024; Gao et al., 2024; Cao et al., 2024) and dilutes environment-driven feedback. Indeed, LLMs carry fundamental risks such as hallucinated outputs, prompt sensitivity, and limited grounding in physical environments (Ji et al., 2023b; Tonmoy et al., 2024; Bang et al., 2025), making their outputs potentially unreliable. Frequent queries also raise scalability concerns due to computational cost and latency (Zhou et al., 2024; Wan et al., 2023). Still, relying solely on RL ignores the structured knowledge encoded in LLMs that could accelerate learning or shape behavior in meaningful ways. *Thus, the fundamental challenge lies in incorporating LLM guidance in a way that leverages its complementary benefits while preserving the optimization dynamics that make RL effective.*

OUR CONTRIBUTIONS. In this work, we propose MIRA (Memory-Integrated Reinforcement Learning Agent), a method that integrates LLM-derived guidance into reinforcement learning through a structured *memory graph*. The memory graph provides an evolving representation of task-relevant information, co-constructed from the agent's experience and LLM outputs. Offline priors initialize the structure, while infrequent online queries on batches of partial observations can further refine it during training. Nodes represent decision-relevant context, such as trajectory segments, and edges encode the hierarchical decomposition linking goals to their subgoals. The graph is designed to remain compact, adding minimal overhead relative to standard replay buffers (Schaul et al., 2015). The memory graph allows the agent to organize and reuse information *without repeated LLM queries*, providing a persistent source of structured knowledge.

Over time, the agent validates, revises, and extends the structure based on its own experience, improving beyond what LLM guidance alone can provide and filtering out mistaken online suggestions. The resulting graph limits dependence on real-time LLM access, alleviating concerns about latency, query cost, and scalability. To integrate the LLM-derived information into learning, we derive a

*utility signal* from the memory graph and use it to softly shape advantage estimates in each RL iteration. This signal guides early rollouts by reinforcing reward-driven gradients when aligned and moderating updates that arise from an inaccurate critic, helping the agent explore more effectively in sparse-reward settings without overriding the environment's feedback. Theoretically, we show that the utility term accelerates early learning. As the policy improves and surpasses the usefulness of LLM-derived guidance, the shaping influence fades, ensuring convergence in the long-horizon limit. We empirically evaluate the effectiveness, sample efficiency, and overhead of incorporating LLM guidance across multiple benchmark environments.

Our contributions are summarized as follows:

- **A MEMORY-INTEGRATED FRAMEWORK FOR LLM GUIDANCE:** We propose **MIRA**, a reinforcement learning agent that integrates LLM-derived guidance through a memory graph co-constructed from agent experience and offline or infrequent online LLM outputs. The graph evolves throughout training, reducing reliance on real-time LLM queries.

- **UTILITY-BASED ADVANTAGE SHAPING:** We introduce **utility-shaped advantage estimation**, which incorporates graph-derived utility into advantage computation without architectural changes and is compatible with any advantage-based policy-gradient method.

- **CONVERGENCE-COMPATIBLE SHAPING:** We provide **theoretical guarantees** showing that by decaying the shaping influence as the policy improves, we preserve long-horizon convergence properties of Proximal Policy Optimization (PPO) (Schulman et al., 2017a) and correct any inaccuracies in LLM outputs.

- **EMPIRICAL VALIDATION ACROSS BENCHMARKS:** We demonstrate **empirically** that MIRA improves sample efficiency over RL and hierarchical baselines and reaches final performance comparable to methods requiring continuous LLM supervision (Zhou et al., 2023; Bhambri et al., 2024), while using far fewer online queries.

The remainder of this paper is organized as follows. Section 2 details MIRA's architecture; Sections 3 and 4 present experimental setup and results across multiple benchmarks; and Section 5 concludes with a discussion of our findings and possible directions for future work.

## 2 METHODOLOGY

We now describe the design of MIRA, whose development is guided by two desiderata: (I) improve early learning by incorporating task-relevant priors from an LLM, (II) minimize reliance on continuous real-time LLM supervision to ensure scalability and maintain autonomous policy learning. MIRA is built on the standard policy-gradient formulation for reinforcement learning (Appendix A).

### 2.1 MEMORY GRAPH DESIGN

The agent maintains an evolving memory graph that organizes information drawn from both LLM suggestions and agent rollouts. Nodes of the graph represent decision-relevant context, and edges encode the hierarchical decomposition of goals into subgoals as provided by the LLM (Figure 2). This structure can be expressed as

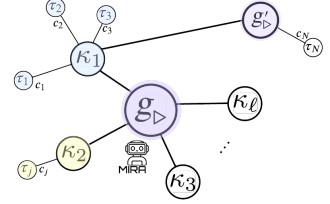

$$\mathcal{G} = \left\{ \left( (o,a)_{\tau_j}, \zeta_j, \hat{r}_j \right)_{c_j} \right\}_{j=1}^{N} \cup \left\{ \kappa_\ell \right\}_{\ell=1}^{L} \cup \left\{ g_\triangleright \right\}. \quad (1)$$

Each trajectory node $j$ consists of a partial observation $o_{\tau_j}$ and an action $a_{\tau_j}$. It is also associated with a goal term $\zeta_j \in \{g_j, \kappa_\ell^{g_j}\}$ indicating either a final goal $(g_j)$ or an abstract subgoal $(\kappa_\ell^{g_j})$ that the trajectory is intended to complete. In addition, the node stores an estimated (sub)goal reward $\hat{r}_j$ for the action sequence, reflecting progress toward completing its associated subgoal, and a confidence score $c_j$ derived from the LLM-derived statistics. The second set of nodes $\{\kappa_\ell\}_{\ell=1}^{L}$ represents subgoals $\kappa_\ell$ provided by the LLM from the environment description. The final term $\{g_\triangleright\}$ denotes the agent's target goal(s).

Figure 2: MIRA's evolving memory graph. Trajectory segments $\tau_j$ are grouped under subgoal nodes $\kappa_\ell$. Subgoals can be shared across multiple final goals, enabling reuse of common behaviors.

The graph is initialized with offline LLM priors and evolves as training progresses. Nodes are added or updated when the agent produces trajectory segments that either introduce a novel segment for a known (sub)goal or achieve a higher estimated return than the existing entry for the same (sub)goal. Existing nodes can also be updated when the agent's experience strengthens entries that were initially derived from low-confidence LLM outputs. Online LLM suggestions may also be added as new nodes when available, provided they pass screening; an online query is triggered only when the rollout utility signal remains near zero for several consecutive episodes, indicating that the current graph offers no helpful guidance. In such cases, the current policy is either exploring regions unsupported by the memory graph or that the graph itself is missing useful segments. Nodes are pruned when they remain unused for a fixed horizon of episodes. Each node maintains an access counter updated whenever it contributes to the utility; nodes whose counters do not change within this window are removed, reflecting reduced relevance to recent rollouts. Offline LLM nodes, though generally stable, may also be removed when rendered obsolete. Additional details about graph construction and environment-specific examples are given in Appendix E. This process keeps the graph compact and adaptive, while ensuring that any low-quality or misleading prior segments naturally fade as higher-return agent-generated trajectories replace them or lead to their pruning.

## 2.2 OFFLINE AND ONLINE GUIDANCE

MIRA incorporates two complementary forms of LLM guidance, accessed either *offline* prior to training or *online* during training. Offline outputs are generated using full access to the task description, providing trajectory segments and subgoal decompositions that initialize the memory graph with structured priors. Offline nodes accelerate early exploration and remain a persistent baseline source of guidance that complements the adaptive updates introduced by online LLM queries.

Online suggestions are incorporated during training when the agent fails to obtain useful guidance (i.e., when the rollout utility remains near zero) from its memory graph for several consecutive episodes. The LLM is constrained to the same partial observability as the agent and, when triggered, may return plans that correspond to short trajectories. Alternatively, it may provide control signals that bias the action preferences over an extended horizon until the current task segment is completed. All online outputs are passed through the *Screening Unit*, which discards low-confidence suggestions. Accepted plans are grafted into the memory graph as new trajectory segments, while accepted control signals bias the policy through soft logit injection, i.e., adding a bounded penalty to the logits of discouraged actions. This penalty induces only a soft preference before the softmax and cannot collapse the action distribution. PPO's clipped objective controls the update size, ensuring that the injected bias functions as lightweight guidance that the critic can override when it strongly disagrees with the value estimates (Biza et al., 2021).

**SCREENING UNIT.** To ensure reliability, online outputs are passed through a lightweight *Screening Unit* designed to reduce hallucinations and reasoning failures (Ji et al., 2023a; Bubeck et al., 2023; Wang et al., 2022; Zhao et al., 2021). Confidence is estimated in two complementary ways. When token-level likelihoods are available, we compute the geometric mean of the per-token probabilities of the completion as a confidence measure. When such likelihoods are unavailable or incomplete (e.g., only top-k likelihoods are provided), we instead estimate confidence by sampling multiple independent completions and measuring agreement across them. A majority-consistency test is then applied to retain only those outputs that appear reliably across the sampled completions. Suggestions that fail to meet a fixed threshold under either criterion are discarded. While this procedure does not eliminate all high-confidence errors, it serves as an effective filter that reduces the risk of hallucinated or low-quality outputs. The screened outputs, referred to as *healthy grafts* in Figure 1, are incorporated into the memory graph as new nodes to further help the policy learning.

## 2.3 UTILITY SIGNAL COMPUTATION

Utility is defined at the level of individual state–action pairs and is computed from the same rollouts used for advantage estimation under the current policy. Each state-action in the trajectory $\tau = \{(o_t, a_t)\}_{t=1}^{T}$ is matched against the corresponding state–action pairs $(o_{t'}, a_{t'})$ in the stored trajectory $\tau_m$. The appropriate memory node $m$ is selected based on the environment instance (e.g., the seed-specific layout) in that training iteration. We then compute the utility for each pair $t$ as:

$$U_t \doteq c_m \cdot \hat{r}_m \cdot \rho(g_{\triangleright}, \zeta_m) \cdot \int \left( (o_t, a_t), (o_{t'}, a_{t'})_{\tau_m} \right). \tag{2}$$

Steps that do not match any stored trajectory segment receive zero utility. The similarity function $f(\cdot, \cdot)$ measures how closely the agent's behavior aligns with the stored trajectory. It incorporates both action agreement and spatial consistency, such as overlap in grid positions or directional alignment in tabular settings (Algorithm 3). To capture semantic structure, the raw similarity score is weighted by a goal-alignment factor $\rho(\cdot, \cdot)$. Each subgoal description specifies a target object or region and a high-level action applied to it; simple rule-based parsing yields a paired entity and action-phase token. The Jaccard similarity between the entity–phase token sets extracted from the agent's target subgoal and each memory entry defines $\rho$. This weighting increases the influence of memory entries that share underlying entities or action phases with the target subgoal and downweights matches corresponding to unrelated parts of the task (Algorithm 4). Thus, a transition contributes to utility only when both its behavioral similarity and its semantic alignment with the relevant subgoal are high. Finally, the score is modulated by the confidence $c_m$ and estimated reward $\hat{r}_m$ attached to the memory node (Algorithm 5). This formulation also helps limit the influence of incorrect LLM guidance that may have passed screening. When an LLM suggestion is inaccurate, its influence naturally diminishes: such segments typically contribute little utility since their estimated reward is low or their similarity score remains small, reflecting weak positional alignment or inconsistent action patterns between the agent's rollout and the stored LLM-derived segment. Further details on the utility computation are presented in Appendix F.

## 2.4 Adaptive Advantage Shaping

We incorporate memory-derived utility into the policy update by augmenting the standard advantage term. Algorithm 1 outlines the shaped PPO update. At iteration $k$, trajectories $\mathcal{D}_k = \{(s_t, a_t, r_t)\}$ are collected under the policy $\pi_{\theta_k}$. The rollout batch is split into minibatches $\mathcal{B}$ for multiple gradient steps. The likelihood ratio $r_t$ compares new and old policies, and the clip parameter $\varepsilon_k$ constrains $r_t$ within $(1 \pm \varepsilon_k)$ as a soft trust region.

The advantage function in policy gradient methods, denoted by $A_t$ at a given time $t$, quantifies how favorable an action $a_t$ is relative to the average action at state $s_t$. It drives learning by reinforcing actions that have higher-than-expected returns and suppressing those that fall short. However, during early training the critic is poorly calibrated due to limited exploration, often producing nearly uniform value estimates across actions (Henderson et al., 2018). As a result, the estimated advantages $A_t$ provide weak learning signals, even when the agent is following behavior that is meaningfully directed toward the task. This issue is particularly pronounced in sparse-reward settings or tasks with delayed feedback, where the critic lacks sufficient signal to distinguish between promising and unproductive behaviors. In such cases, the estimated advantage tends to be near-zero or highly noisy for most timesteps, especially early in training (Figure 15).

---

**Algorithm 1** Shaped PPO actor (changes)

> **for** $k = 0, 1, \ldots$ **do**
>  Collect $\mathcal{D}_k = \{(s_t, a_t, r_t)\}$ using $\pi_{\theta_k}$
>  Compute $A_t$ and $U_t$ from rollouts
>  $\tilde{A}_t = \eta_t A_t + \xi_t U_t$
>  **for** $epoch = 1$ to $K$ **do**
>   **for** minibatch $\mathcal{B} \subset \mathcal{D}_k$ **do**
>    $r_t(\theta) = \pi_\theta(a_t|s_t)/\pi_{\theta_k}(a_t|s_t)$
>    $\mathcal{L}^{\text{shaped}}(\pi_\theta) = \mathbb{E}\left[\min(r_t, 1 \pm \varepsilon_k)\tilde{A}_t\right]$
>    $\theta \leftarrow \theta + \alpha_\theta \nabla_\theta \mathcal{L}^{\text{shaped}}(\pi_\theta)$
>   **end for**
>  **end for**
> **end for**

---

To address this, we introduce a shaped advantage as:

$$\tilde{A}_t = \eta_t A_t + \xi_t U_t, \quad 0 < \eta_t \leq 1, \; \xi_t \leq \delta \eta_t, \; \delta \in [0, 1), \; \lim_{t \to \infty} \eta_t = 1, \; \lim_{t \to \infty} \xi_t = 0. \quad (3)$$

This formulation preserves the fundamental role of the advantage function, while refining it with utility-based guidance. It forms a cooperative process between critic predictions and the memory-derived utility. The critic provides an estimate based on learned reward prediction and bootstrapping, while the utility term injects an inductive bias derived from language-guided priors. Together, they form a joint estimator in which each component compensates for the other's limitations without distorting policy optimization. When the critic signal is weak due to insufficient value discrimination, the resulting gradients are uninformative and impair the agent's ability to bootstrap from sparse or delayed rewards. The utility term provides additional directional guidance aligned with task objectives, accelerating learning by compensating for weak or flat gradients (Figure 15, Appendix G).

As training progresses and the policy becomes more accurate, the critic's advantage estimates $A_t$ become more reliable. Accordingly, the shaping weight $\xi_t$ is annealed to reduce the direct influence of the utility term and $\eta_t$ is ramped toward 1 over training. Since the LLM-derived signals can be imperfect, annealing $\xi_t$ prevents inaccuracies in those signals from being preserved in the asymptotic regime. This ensures that the final policy is optimized with respect to the true reward function $\mathcal{R}$ and remains consistent with PPO's stability guarantees. Early in training, however, $\xi_t$ remains large enough for the utility to accelerate exploration using the LLM's possibly imperfect, but still useful, prior knowledge. As $\xi_t$ decays, any suboptimalities in these priors are naturally learned away, yielding the benefits of shaping during the sparse-reward phase, without biasing long-run behavior or altering the policy or critic structure. See remark C.4 for more explanation.

Before turning to experiments, we provide an interpretive perspective on how adaptive advantage shaping affects optimization in sparse-reward regimes and in Appendix establish that the proposed shaping mechanism preserves the policy improvement property of PPO under standard boundedness and scaling assumptions, which we formally enumerate in Appendix C.1. More broadly, the method remains compatible with policy gradient algorithms that relies on advantage estimation, offering a general mechanism for integrating language-derived priors into RL.

**Theorem 1** (Non-Vanishing Updates in Sparse-Reward Regimes). *Define the shaped surrogate* $\mathcal{L}^{shaped}(\theta) \doteq \mathbb{E}\left[\nabla_\theta \log \pi_\theta(a_t|s_t)\,\tilde{A}_t\right]$, *and the PPO surrogate* $\mathcal{L}^{ppo}(\pi) = \mathbb{E}[\nabla_\theta \log \pi_\theta(a_t|s_t)\,A_t]$. *Consider a training iteration $k$ such that the expected magnitude of the PPO advantage is small, i.e., $\mathbb{E}[|A_t|] \leq \varepsilon_A$ for some $\varepsilon_A \approx 0$. Under Assumptions 1–3, the expected norm of the shaped PPO policy update satisfies*

$$\left\|\mathcal{L}_k^{\text{shaped}}\right\| \;\geq\; \xi_k \left\|\mathcal{L}_k^U\right\| - O(\varepsilon_A),$$

*where* $\mathcal{L}_k^U \doteq \mathbb{E}[\nabla_\theta \log \pi_\theta(a_t|s_t)\,U_t]$.

*Proof.* Deferred to Appendix C.

## 3 EXPERIMENTAL SETUP

We validate our method through extensive experiments implemented using the RLlib (Liang et al., 2018). Our evaluation focuses on performance gains, sample efficiency, and the computational overhead introduced by LLM integration. The objective is to characterize the benefits and trade-offs of incorporating LLM guidance in RL, including how different levels of LLM capabilities influence the policy learning dynamics and final policy quality.

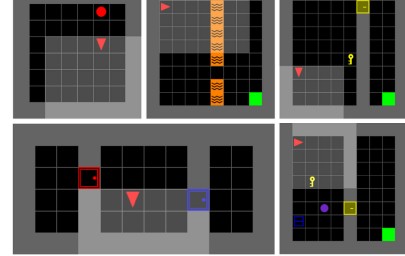

Figure 3: Evaluation environments. Top: REDBALL (navigation to target), LAVACROSSING (long-horizon navigation with irreversible hazards), DOORKEY (sparse reward with key–goal dependency). Bottom: REDBLUEDOOR (sequence-sensitive toggling), DISTRACTED DOORKEY (distractor-rich variant with key-goal dependency).

### 3.1 SIMULATION PLATFORM

We consider six distinct environments, which are selected to span discrete vs. visual inputs, short- vs. long-horizon dependencies, reversible vs. irreversible dynamics, and with vs. without perceptual distractors, forming a compact yet representative benchmark for sparse-reward RL.

GYMNASIUM TOYTEXT. Gymnasium (Arnoldo et al., 2024) provides simple tabular environments for controlled analysis of learning dynamics in low-dimensional settings. Despite their simplicity, these environments feature sparse rewards and require strategic exploration, making them suitable for isolating the early-stage benefits of memory-guided utility shaping. We include FROZENLAKE as a minimal benchmark where PPO reliably converges to the optimal policy, enabling us to verify that MIRA preserves convergence while accelerating early learning.

MINIGRID AND BABYAI. MiniGrid (Chevalier-Boisvert et al., 2023) and BabyAI (Chevalier-Boisvert et al., 2019) are suites of lightweight, procedurally generated environments designed to

evaluate exploration and planning in partially observable, sparse-reward settings. We use these tasks to assess the effectiveness of advantage shaping in long-horizon decision-making environments that require reasoning under uncertainty and robustness to irrelevant stimuli. We include five tasks, selected to cover diverse challenges involving planning, credit assignment, and distraction resilience (Figure 3). We use pixel-based observations (RGB images) rendered from the environment as the policy inputs, to introduce perceptual complexity and evaluate agent performance under a more realistic observation setting.

## 3.2 BASELINE METHODS

PPO (RL BASELINE). We train a tabula rasa PPO agent (Schulman et al., 2017a) that learns purely from environment interaction and rewards. Network architecture, PPO hyperparameters, and rollout settings are held fixed across all methods for fair comparison.

HIERARCHICAL RL. We include hierarchical reinforcement learning (HRL) (Matthews et al., 2022) as a baseline that uses pre-trained LLM option policies for temporal abstraction.

LLM-RS. We consider the method of (Bhambri et al., 2024), which we refer to as LLM-RS. This approach queries the LLM in real time to generate plans for potential-based reward shaping, with a verifier refining them for valid action sequences.

LLM4TEACH. We include LLM4Teach (Zhou et al., 2023) as a representative teacher-based approach. It employs a pre-trained LLM as a policy teacher and guides the RL agent through policy distillation, and is among the state-of-the-art methods in this category.

## 4 EXPERIMENTAL RESULTS

To assess how effectively MIRA addresses the challenges raised in the introduction, we structure our core experiments around three guiding questions. These questions examine whether utility-shaped advantages compensate for PPO's weak early gradients in sparse-reward, long-horizon, and partially observable settings, and whether amortizing limited LLM guidance into a persistent memory structure provides sufficient return while supporting scalability and avoiding latency. Formally:

- **Q1**: How does MIRA improve early learning efficiency and convergence relative to PPO, even when PPO's final performance is competitive?

- **Q2**: How well does MIRA perform in long-horizon environments that demand extended exploration and multi-step reasoning?

- **Q3**: How effectively does MIRA translate a limited number of LLM queries into performance gains compared to query-heavy methods?

To isolate the contribution of individual components in MIRA's design, we further conduct ablations organized around three additional questions. These probe how online queries complement offline-initialized memory, and how sensitive MIRA is to degraded, incorrect, or stylistically varied LLM outputs. Formally:

- **Q4**: How do online LLM queries improve learning, beyond what offline memory provides?

- **Q5**: How does MIRA handle late-stage exposure to degraded LLM guidance once its memory is well-formed?

- **Q6**: How do variations in LLM reasoning affect memory and downstream results?

Appendix G provides additional results, including evaluations on unseen seeds to assess generalization, wall-clock analyses of LLM-query overhead (Figure 16), and measurements of memory growth (Figure 17). We also report ablations on the screening threshold (Figure 14) and prompt wording (Figure 13), each examining its effect on overall learning behavior and addressing aspects of Q5. In addition, supplementary plots from sweeps over shaping weights (Figure 15) are included to analyze their impact on early-stage learning dynamics and reward progression.

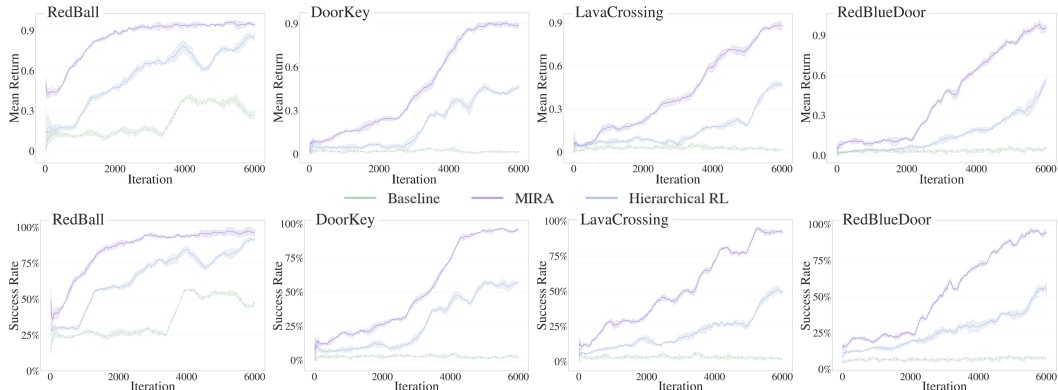

Figure 5: Mean return (top) and success rate (bottom) across four MiniGrid and BabyAI tasks. MIRA consistently outperforms both baselines, achieving faster learning, higher asymptotic return, and greater success rates. These results are obtained with a small LLM budget, using fewer than ten offline prompts to build memory graphs plus infrequent online queries to guide exploration.

## 4.1 TABULAR BENCHMARK AND PARTIALLY OBSERVABLE TASKS

**Q1:** We evaluate two variants of MIRA on FROZENLAKE-8X8, an offline-only version and an online-only version, and compare them to the PPO baseline, averaging results over four seeds. In the offline variant, three zero-shot GPT-o4-mini queries generate an initial memory graph. The LLM observes the grid layout (matching the agent's full observability) but does not receive the slipperiness probability, which is hidden from both. As shown in Figure 4, this initialization results in faster early learning, and the offline variant maintains a higher return than PPO throughout, and the online-only variant in the first 1K iterations. The online-only variant begins with an empty memory graph, without any global information about the map and issues LLM queries during training.

Since the layout is deterministic and the action set is small, these online queries can infer short segments from the agent's rollouts and populate the memory graph as training progresses. This provides a faster improvement rate than the offline variant, though it requires more LLM queries. PPO eventually catches up, and by convergence, the asymptotic returns of all three methods become comparable. During training, the shaping signal primarily affects the early iterations. As the policy improves, $\eta$ increases, $\xi_t$ decreases, i.e. the shaping ratio $\delta_t = \xi_t/\eta_t$ decays, reducing reliance on memory. Under standard stochastic approximation conditions (Kushner & Yin, 2003), this decay keeps the critic error within an $O(\delta_t)$ neighborhood of the true value, which vanishes as $\delta_t \to 0$.

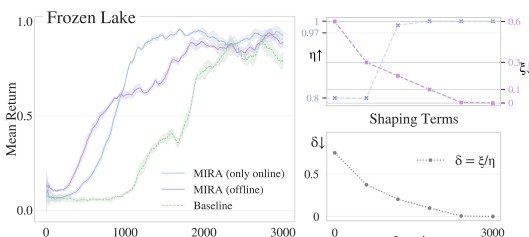

Figure 4: Mean return on FROZENLAKE (left): Both MIRA variants improve early-stage learning relative to PPO, while PPO eventually attains a comparable asymptotic return. Evolution of shaping terms $\eta_t$, $\xi_t$, and ratio $\delta_t$ (right): $\delta_t$ decays during training, ensuring convergence as $\delta_t \to 0$.

**Q2:** We next evaluate MIRA on five tasks designed to isolate distinct challenges in sparse and partially observable environments. Figure 5 shows mean return and success rate across the four tasks, with performance averaged over four different seeds. In simpler tasks such as REDBALL, PPO shows moderate early gains but plateaus well below optimal performance. Although hierarchical RL eventually catches up, MIRA reaches optimal returns in under half the training iterations. In LAVACROSSING, PPO fails to improve beyond near-zero success, indicating ineffective exploration. Hierarchical RL improves steadily but converges more slowly than MIRA. In more complex tasks such as DOORKEY and REDBLUEDOOR, MIRA achieves substantially higher success rates, approximately twice those of HRL, while also converging faster.

These gains are achieved with a limited LLM query budget that combines offline and infrequent online access. Offline queries scale with task complexity. In REDBALL, four zero-shot prompts to GPT-o4-mini are sufficient to build a useful memory graph, while DOORKEY requires about seven

queries that mix few-shot and zero-shot prompts. Online queries are budgeted separately and also vary with task complexity. In REDBALL, about seven online queries suffice to suppress irrelevant actions throughout training. In REDBLUEDOOR, queries are triggered more frequently early in training to help interpret partial observations and suggest short sequences, such as turning, that align the agent with the door. Once the red door is discovered and toggled, the offline memory becomes sufficient. In this task, rooms behind the doors serve only as distractors; baseline agents, including hierarchical RL, often waste steps exploring them. As shown in Figure 5 (lower right), HRL achieves higher success rates than PPO but yields similar average return in the beginning due to suboptimal trajectory use. By contrast, MIRA avoids such inefficiencies by focusing on goal-relevant behavior earlier in training.

**Q3:** To further evaluate MIRA, we compare it to LLM4Teach and LLM-RS in the custom variant environment DISTRACTED DOORKEY. We also include a Sole LLM baseline, where GPT-o4-mini executes plans under full observability without learning. Figure 6 shows mean return progression at selected training checkpoints. For Sole LLM, we report average return over 100 seeds to demonstrate that the task is LLM-solvable and that its outputs provide useful structural guidance. The accompanying bar chart reports amortized return per cumulative LLM query under two fixed budgets (10 and 20 queries, both below the usage of any method), quantifying how each method translates queries into performance gains.

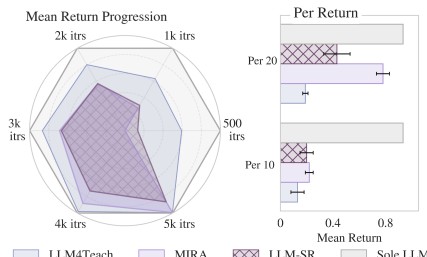

Figure 6: Mean return (left): LLM4Teach shows faster early gains, while MIRA steadily improves and matches its final return with fewer queries. LLM-RS benefits early from reward shaping but plateaus lower. Return per LLM query (right): Under two query budgets, MIRA achieves higher efficiency.

MIRA achieves higher query efficiency than both LLM4Teach and LLM-RS, using about 30 queries per run (seven offline and $20 \pm 3$ online) to obtain higher return per query. In contrast, LLM4Teach issues dense supervision, querying the LLM once on every state–action–reward triplet within training batches, which in our experimental setup corresponds to more than 500 queries before the policy stabilizes. LLM-RS, which uses LLM-generated reward code, queries once per layout, totaling over 50 calls in our setup. While lighter than LLM4Teach, this still requires layout-level access throughout training. Despite its heavier budget, LLM4Teach achieves comparable final performance to MIRA, while LLM-RS fails to match MIRA's return. Notably, LLM-RS outpaces MIRA early due to reward shaping, but falls behind later. LLM4Teach shows an early advantage through front-loaded queries, but at a significantly higher cost. Tables in Appendix G report results on unseen evaluation seeds to assess generalization.

## 4.2 ABLATION STUDIES

**Q4** (ONLINE QUERY FREQUENCY)**:** We vary the number of online LLM queries issued during training of DOORKEY, to assess how constrained usage affects learning efficiency and final performance. Each agent begins with the same memory graph, initialized from identical offline queries, isolating the contribution of dynamic LLM input from that of static memory initialization. We compare MIRA under three online budgets: zero, a mid budget of 10 queries, and a high budget of 20. As shown in Figure 7 (left), more frequent online access accelerates learning, with the large-budget variant achieving optimal return in fewer steps (Table 1, Appendix G). Even with just 10 online calls, MIRA substantially outperforms the offline-only variant. Nevertheless, MIRA (offline) still yields a notable boost over baseline PPO, indicating that static memory alone can provide meaningful guidance when dynamic access is unavailable. Given this noticeable improvement over PPO, the offline-only variant is a practical choice when run-time LLM access is restricted or when latency makes online queries impractical, since it relies entirely on the memory graph constructed offline. When limited run-time access is available, the online variant becomes preferable because dynamic queries update the memory during exploration and accelerate learning, at the cost of additional wall-clock time associated with LLM calls (Figure 16, Appendix G).

**Q5** (UNRELIABLE LLM OUTPUTS)**:** We evaluate a scenario where the LLM is swapped at a later training stage and the screening unit is disabled only for this final online phase in DISTRACTED

Figure 7: **Ablation Studies**. Query frequency (left): Agents share the same offline memory but vary in online budgets. More queries accelerate learning, with high-budget agents achieving optimal returns more quickly. Unreliable LLM (middle): With identical offline memory, screening is disabled and queries are swapped from GPT-o4-mini to Gemini Pro only in the late phase. Performance remains stable in the late phase, indicating reduced dependence on online guidance once policy have matured. LLM models (right): Agents differ only in the LLM used for memory. Performance differences reflect divergent reasoning styles: Gemma3 induces inefficient checking, Claude favors cautious exploration, while Gemini Pro and o4-mini enable faster learning.

DOORKEY environment. All agents share the same offline-initialized memory graph and use GPT-o4-mini with screening during earlier online queries. In the final stage, we replace the LLM with Gemini Pro and omit screening. By this point, MIRA has accumulated sufficient experience and memory, allowing it to tolerate low-confidence or incorrect suggestions without collapsing performance. We prompted both LLMs with a scenario where the agent has already explored *thoroughly* and confirmed no key is present (implying it was already collected, since inventory is *unobserved*). When asked which action to down-weight, GPT-o4-mini gave a consistent suppression, whereas Gemini returned a misaligned alternative. As shown in Figure 7 (middle), MIRA remains stable under degraded guidance, though convergence slows and final return drops slightly. Details of the LLM responses are shown in Figure 12, Appendix D.

**Q6** (REASONING AND PERFORMANCE): We assess MIRA's sensitivity to the choice of language model by replacing GPT-o4-mini with alternatives such as GPT-4o (OpenAI, 2024), Claude Sonnet 4 (Anthropic, 2024), Gemma 3 27B (Ananthanarayanan et al., 2024), Gemini 2.5 Flash and Pro (Chen et al., 2024). All models go through the same process to ensure comparability. Unlike the ablation done before, the model swap is applied from the beginning of training. As shown in Figure 7 (right), the reasoning style shaping the memory graph strongly impacts downstream RL performance. For instance, Gemma3 performs poorly because it recommends checking the door after every pickup, leading to wasteful steps. Claude adopts an exploratory policy that yields slow but eventual progress, showing early improvement followed by plateauing, but it eventually recovers as the memory is dynamic. GeminiPro and GPT-o4-mini both enable fast early learning, but o4-mini's memory includes *detours* that prove beneficial later, ultimately reaching the highest asymptotic return. These differences highlight how the reasoning processes behind LLM outputs directly influence MIRA's long-term policy quality. Figure 11 in Appendix D presents the reasoning traces produced by different LLM models used.

## 5 CONCLUSION AND FUTURE WORK

We propose MIRA, a reinforcement learning (RL) framework that integrates large language model (LLM) guidance via a memory graph built from high-return trajectories and LLM-inferred information. By shaping advantage estimates with a utility signal derived from this memory, MIRA accelerates early learning without requiring continuous LLM supervision. Theoretical and empirical results on sparse-reward tasks confirm improved sample efficiency and preserved convergence. Limitations of the current design are discussed in Appendix H. Extending MIRA to continuous-control and vision-based domains is a natural next step. Prior LLM-guided robotics work shows that semantic subgoals can be grounded in continuous spaces through learned affordances or visual embeddings (Brohan et al., 2023; Huang et al., 2022). In such settings, MIRA's discrete similarity function can be replaced with comparisons in a latent representation space, e.g., using encoder features or R3M-style embeddings (Nair et al., 2022), while memory scalability can be maintained through clustered or hierarchical organization. These adaptations would allow MIRA's memory graph and utility-based shaping to apply beyond grid worlds and support longer-horizon tasks with richer state spaces. Extending MIRA to multi-goal domains such as Crafter (Hafner et al., 2023), where reusable subgoal structure is prominent, is another promising direction.

## ACKNOWLEDGMENTS

This work was supported in part by the National Science Foundation (NSF) under grants CNS-2533813 and CNS-2312761 and the Office of Naval Research under grant N000142412073.

## REPRODUCIBILITY STATEMENT

We have taken several steps to ensure reproducibility of our results. All theoretical assumptions and complete proofs are included in Appendix C. Appendix D details the environment specifications and the exact LLM prompts used for both offline and online queries. Appendix I lists the full set of hyperparameters for MIRA across every evaluated environment. We also provide pseudocodes for all proposed algorithms in Algorithms 1 through 5, ensuring clarity and transparency despite their straightforward implementation. The code is available at paper's webpage `https://narjesno.github.io/MIRA/`. Together, these materials supply all information necessary to reproduce our experiments and verify the claims of the paper.

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

APPENDIX

The supplemental material is organized as follows:

## A  BACKGROUND

### A.1  STANDARD REINFORCEMENT LEARNING

Reinforcement learning (RL) is typically modeled as a Markov decision process (MDP), defined by a tuple $(\mathcal{S}, \mathcal{A}, P, r, \gamma)$, where $\mathcal{S}$ is the state space, $\mathcal{A}$ the action space, $P$ the transition function, $r$ the reward function, and $\gamma \in [0, 1)$ the discount factor. The agent's behavior is determined by a policy $\pi$, which defines a probability distribution over actions given the current state: $\pi(a|s)$. Learning proceeds through interaction with the environment, producing trajectories, sequences of states, actions, and rewards of the form $\tau = (s_0, a_0, r_0, s_1, a_1, r_1, \dots)$, and using them to improve the policy.

The objective is to learn a policy that maximizes the expected return, defined as the discounted sum of rewards along a trajectory:

$$\mathbb{E}_\pi \left[ \sum_{t=0}^{\infty} \gamma^t r(s_t, a_t) \right]. \tag{4}$$

The environment's reward function implicitly defines the final goal ($g_\triangleright$) by assigning reward to behaviors that accomplish the task (Sutton et al., 1998; Silver, 2015). To estimate this objective, RL algorithms often make use of value functions, which quantify the long-term utility of states or state-action pairs. The state-value function $V(s)$ denotes the expected return when starting from state $s$ and following policy $\pi$:

$$V(s) = \mathbb{E}_\pi \left[ \sum_{t=0}^{\infty} \gamma^t r(s_t, a_t) \mid s_0 = s \right]. \tag{5}$$

The action-value function $Q(s, a)$ further conditions on the first action taken and is defined as:

$$Q(s, a) = \mathbb{E}_\pi \left[ \sum_{t=0}^{\infty} \gamma^t r(s_t, a_t) \mid s_0 = s, a_0 = a \right]. \tag{6}$$

### A.1.1 Partial Observability and Credit Assignment Challenges

In many real-world scenarios, the environment is only partially observable. In such cases, the MDP generalizes to a partially observable MDP (POMDP), defined by the tuple $(\mathcal{S}, \mathcal{A}, P, r, \gamma, \mathcal{O}, \Omega)$, where $\mathcal{O}$ is the observation space and $\Omega$ is the observation function. The agent does not directly observe the true state $s_t \in \mathcal{S}$; instead, it receives observations $o_t$ from an observation space $\mathcal{O}$, sampled via $\Omega(o_t|s_t)$, and must rely on its history of observations and actions to make decisions (Kaelbling et al., 1998).

These difficulties are further amplified in environments where the agents face sparse and delayed rewards. Sparse rewards refer to the limited presence of nonzero rewards since this feedback is only provided upon reaching specific goals (i.e., $r(s_t, a_t)$ is typically zero until the agent reaches the final goal state ($g_\triangleright$) defined by the task). On the other hand, delayed rewards refer to settings where the consequences of an action are not reflected in the reward until several steps later. In both cases, the agent must reason over long horizons to determine which actions contributed to the eventual outcome, a challenge known as the credit assignment problem (Schulman et al., 2015a).

Credit assignment is closely tied to the broader challenge of exploration. Inefficient exploration occurs when the agent fails to sufficiently cover the state space, limiting its ability to discover high-return trajectories and improve its policy. This problem is exacerbated in high-dimensional environments, where the number of possible state-action sequences grows exponentially and random exploration becomes increasingly unlikely to encounter informative transitions with sparse or delayed rewards. In such cases, the combination of large search spaces and limited reward signals often leads to slow convergence, poor sample efficiency, and high variance in learning outcomes.

### A.1.2 Subgoals and Abstractions

In long-horizon tasks, reinforcement learning agents often benefit from structuring behavior around subgoals, intermediate objectives that facilitate progress toward the overall task. The concept of subgoals in reinforcement learning originated in hierarchical reinforcement learning (HRL), where it was formalized through the use of temporally extended actions. In particular, the options framework introduced by Sutton et al. (1999) defines options as high-level actions composed of an initiation set, a policy, and a termination condition, often interpreted as achieving a subgoal (Stolle & Precup, 2002). These subgoals correspond to intermediate states or conditions that decompose long-horizon tasks into smaller, temporally coherent segments that make the final goal more attainable when reached. More broadly, subgoals provide structure for reasoning over extended time horizons and facilitate learning in sparse-reward settings.

While early approaches focused on explicit or learned state-based subgoals, recent work has explored abstract subgoals that capture semantic or latent-level progress. These abstractions may not correspond to a specific state but instead reflect high-level intentions and meaningful progress (e.g., opening a door, entering a room, or collecting an object). Such abstractions enable reasoning at a higher level of granularity and are especially useful in environments with sparse rewards or delayed feedback. Subgoal discovery and abstraction have also been explored in curriculum learning, imitation learning, and human-in-the-loop frameworks to improve exploration and sample efficiency (MacGlashan et al., 2017; Shiarlis et al., 2018; Narvekar et al., 2020).

### A.2 Policy Gradient Methods

Policy gradient methods directly optimize a parameterized policy $\pi_\theta(a|s)$ by ascending the gradient of expected return. The objective is to find parameters $\theta$ that maximize:

$$J(\theta) = \mathbb{E}_{\tau \sim \pi_\theta} \left[ \sum_{t=0}^{\infty} \gamma^t r(s_t, a_t) \right], \tag{7}$$

where $\tau$ denotes a trajectory generated by following the current policy. The gradient of this objective can be estimated via the likelihood ratio trick, yielding the REINFORCE estimator (Williams, 1992):

$$\nabla_\theta J(\pi_\theta) = \mathbb{E}_\pi \left[ \nabla_\theta \log \pi_\theta(a_t|s_t) R \right], \tag{8}$$

where $R_t$ is the return from time $t$ onward. While theoretically sound and unbiased, this estimator suffers from high variance, making it challenging to apply in practice without further refinement.

### A.2.1 ADVANTAGE-BASED POLICY OPTIMIZATION

To reduce variance and improve sample efficiency, modern policy gradient algorithms often use advantage functions, which quantify the relative quality of an action compared to the policy's baseline behavior. The advantage function is defined as:

$$A(s,a) = Q(s,a) - V(s), \tag{9}$$

where $Q(s,a)$ is the expected return from taking action $a$ in state $s$, and $V(s)$ is the expected return from $s$ under policy $\pi$. Using this formulation, the policy gradient becomes:

$$\nabla_\theta J(\pi_\theta) = \mathbb{E}_\pi \left[ \nabla_\theta \log \pi_\theta(a_t \mid s_t) A \right], \tag{10}$$

which improves stability while preserving unbiasedness.

This idea underpins a family of actor-critic algorithms, where the actor updates the policy using the advantage-weighted gradient, and the critic estimates value functions used to compute $A(s,a)$. Representative algorithms in this class include A2C and A3C (Mnih et al., 2016), which leverage parallel actors to accelerate training and stabilize updates, and PPO (Schulman et al., 2017b), which constrains policy updates by clipping the policy ratio in the surrogate objective:

$$\mathcal{L}^{\text{PPO}}(\pi) = \mathbb{E} \left[ \min(r_t A_t, \text{clip}(r_t, 1 - \varepsilon, 1 + \varepsilon) A_t) \right], \tag{11}$$

where $\varepsilon > 0$ is a small trust region parameter that limits how much the policy is allowed to change at each update.

These methods are widely used in modern deep reinforcement learning due to their scalability and consistent empirical performance across a range of tasks. Since MIRA operates by shaping the advantage function, it is compatible with any policy optimization method that relies on advantage-weighted updates.

## B RELATED WORKS

### B.1 LANGUAGE MODEL GUIDANCE IN RL

A growing line of work explores how large language models (LLMs) can be integrated into reinforcement learning by framing them as auxiliary components within the agent–environment loop. A recent taxonomy by Cao et al. (2024) outlines the roles of LLMs in RL along four key dimensions: information processors, reward designers, decision-makers, and generators.

As *information processors*, LLMs extract and organize task-relevant knowledge from natural language, environment descriptions, or prior experience. This includes synthesizing high-level goals, parsing instructions, and transforming language input into actionable constraints or representations (Wang et al., 2024a; Shinn et al., 2023). A common approach is to use frozen pre-trained models to encode task-relevant features without fine-tuning, though they may perform poorly on out-of-distribution data due to limited adaptability (Radford et al., 2021; Paischer et al., 2022; 2023). Alternatively, fine-tuned models can better align with task-specific distributions, leading to more robust RL performance and improved generalization in unseen environments (Kim et al., 2023; Poudel et al., 2024). In addition, LLMs can convert human instructions or task prompts into formal representations or structured goals, and interpret descriptions of the environment, such as objects, layouts, or dynamics, into usable priors for downstream RL modules. This reduces the burden of language comprehension for RL agents and improves sample efficiency (Basavatia et al., 2024; Sumers et al., 2021; Song et al., 2023a; Liang et al., 2022). These models can decouple information processing from control, with the LLM handling language grounding and feature extraction while the policy module focuses on decision-making. Such capabilities can reduce learning complexity and accelerate policy acquisition by shaping the agent's representation space early in training.

As *reward designers*, LLMs provide auxiliary supervision by scoring agent behavior or generating rewards. This can take the form of natural language critiques, programmatic reward code, or goal-conditioned evaluations. In the implicit reward setting, LLMs serve as proxy reward models

by either being directly prompted to evaluate agent behavior (Chu et al., 2023; Wu et al., 2023) or by computing alignment between visual observations and language goals using pretrained vision-language models (Wang et al., 2024b; Adeniji et al., 2023; Seo et al., 2023; Grauman et al., 2022). These methods enable reward shaping via natural language instructions or preference feedback and have been shown to improve learning efficiency and generalization. In the explicit reward setting, LLMs are used to generate executable code that defines reward functions programmatically. This includes frameworks that iteratively refine reward code using self-reflection and feedback from training outcomes (Yu et al., 2023; Madaan et al., 2023; Song et al., 2023b). Compared to manually engineered rewards, these LLM-generated functions offer transparency and adaptability, and in some cases match or exceed human performance, especially in complex manipulation tasks.

As *decision-makers*, LLMs output action plans, policy sketches, or even direct action sequences based on current observations. These methods embed LLMs tightly into the decision loop, either guiding exploration or dictating behavior in few-shot or zero-shot settings. One approach leverages pre-trained LLMs for direct action generation, often adapting transformer-based models like Decision Transformers to treat offline RL as a sequence modeling problem. These LLM-backed policies show improved generalization, particularly in sparse-reward and long-horizon tasks, by transferring latent structure learned from large-scale language data. Some methods further fine-tune LLMs using task-specific trajectories or append small task-specific modules to facilitate adaptation, achieving notable gains in sample efficiency and task transfer (Zitkovich et al., 2023; Shi et al., 2023; Mezghani et al., 2023).
Other works integrate LLMs more loosely as action guides, generating action candidates or expert priors to support exploration and training. For example, LLMs can prune the action space by proposing high-probability candidates or decompose complex goals into sequential subtasks, improving exploration in environments with large or unstructured action spaces (Yao et al., 2020; Hausknecht et al., 2020; Dalal et al., 2024; Wan et al., 2025). They have also been used to regularize policy updates, align agent behavior with human intent, or inject expert-level motion plans. Across both low-level and strategic roles, LLM-based decision-making enables agents to learn from rich, structured priors and execute more informed behaviors in complex settings.

As *generators*, LLMs contribute to reinforcement learning by either simulating environmental dynamics or providing policy-level explanations to enhance transparency. In the simulation role, LLMs function as world model simulators that generate trajectories or learn latent dynamics representations from multimodal data, thereby improving sample efficiency in model-based RL. Recent work has leveraged Transformer-based architectures to model complex visual or sequential environments, demonstrating gains in generalization and long-horizon reasoning. These models either auto-regressively generate rollouts from pre-trained dynamics or use representation learning to predict future states and rewards, often incorporating language as an additional modality for grounding and abstraction (Micheli et al., 2022; Chen et al., 2022; Robine et al., 2023). Separately, LLMs have been used as policy interpreters to generate human-readable explanations of agent behavior from state-action histories or decision trees. This facilitates interpretability, improves human trust, and can inform reward design or debugging, though current work has focused mainly on policy-level summaries (Lin et al., 2023; Lu et al., 2023).

While MIRA incorporates elements of information processing and LLMs as generators, its overall orientation remains distinct and more RL-centric from prior LLM-centered approaches. Rather than positioning the LLM as a decision-maker or continuous feedback provider, MIRA relegates it to a supporting role that gradually fades over time. LLM outputs are used intermittently to enrich a structured memory graph that informs, but does not dictate, learning. The primary learning signal remains grounded in environment interaction, with utility shaping softly modulating advantage estimates rather than overriding the reward function. This design prioritizes policy optimization through reinforcement learning rather than imitation or prompting

## B.2 Memory and Buffers in RL

Augmenting RL agents with structured memory has been proposed as a means of supporting generalization, planning, and long-horizon credit assignment. Early works such as Neural Episodic Control (NEC) and other episodic value-based methods enabled agents to recall high-value past experiences for more sample-efficient decision-making via memory buffers (Pritzel et al., 2017; Blundell et al., 2016; Lin, 1992). Subsequent approaches extended this idea by integrating differentiable memory

into policy networks (Qiu et al., 2024). Other methods introduce structured representations, such as subgoal graphs or navigation maps, to facilitate hierarchical planning, exploration, or navigation in partially observable environments (Beeching et al., 2020; Rana et al., 2023). Across these directions, the common pattern is to directly query stored structures, either through replay, imitation, or graph traversal, to guide behavior.

MIRA aligns with this direction by maintaining a structured memory graph populated with high-return trajectory segments but departs from this pattern in several key ways. First, its memory graph is co-constructed from high-return agent trajectories and LLM-inferred subgoals, enabling abstraction and structure difficult to obtain early through interaction alone. Second, rather than querying memory for action selection or value estimation, MIRA distills the stored information into a utility signal that modulates advantage estimates during training. This indirect shaping avoids disrupting the optimization loop or overfitting to specific stored transitions. Finally, MIRA maintains a compact memory via pruning and infrequent updates, which avoids the inefficiencies of excessive memory or the brittleness of sparse guidance (Liu & Zou, 2018). This makes MIRA more scalable and better suited for tasks where long-term structure must complement autonomous learning.

### B.3 ADVANTAGE MODIFICATIONS IN RL

Modifying the advantage function has been studied as a way to stabilize learning and improve sample efficiency in policy optimization. A common approach adjusts the estimation process to better balance bias and variance. Generalized Advantage Estimation (GAE) (Schulman et al., 2015b) introduces a tunable parameter that interpolates between high-bias low-variance and low-bias high-variance estimators, and is widely adopted in actor-critic algorithms. Other methods reformulate policy updates in terms of advantages. Advantage-Weighted Regression (AWR) (Peng et al., 2019) avoids policy gradients and instead performs weighted regression over actions. P3O (Fakoor et al., 2020) combines on-policy and off-policy learning by applying advantage-weighted importance sampling to stabilize updates. In the offline RL setting, advantage estimates are often used to filter experience and address distributional shift. Advantage-based data selection (Kostrikov et al., 2021) discards transitions with low advantage, helping to focus learning on high-quality samples. Additional work incorporates auxiliary signals into the advantage estimate. Preference-based RL (Lee et al., 2021) derives implicit advantage signals from human comparisons, while other approaches integrate value correction from ensemble critics or confidence measures to adjust learning.

MIRA builds on these ideas but takes a different path. Instead of replacing the estimator or introducing new objectives, it shapes the advantage using a utility term derived from a structured memory graph. This utility reflects agent experience and LLM-derived subgoals, allowing guidance without overriding reward feedback. The resulting signal is integrated into PPO's update rule without disrupting its optimization dynamics, enabling structured shaping while maintaining scalability and convergence guarantees.

## C  THEORETICAL RESULTS

Since the utility term does not alter the policy or critic structure, and enters additively, MIRA preserves standard stability and asymptotic properties of policy gradient methods such as PPO under standard assumptions:

### C.1  ASSUMPTIONS

**Assumption 1** (Boundedness).

    *a. For all updates $k$ and all $(s, a)$*

$$|A_k(s,a)| \leq A_{\max}, \quad 0 \leq U_k(s,a) \leq U_{\max} \tag{12}$$

    *b. Define the scale-adjusted shaping term as:*

$$U_k(s,a) = \bar{A}_k \cdot U_k(s,a), \quad where \ \bar{A}_k = \langle |A_k| \rangle \tag{13}$$

    *and set*

$$U_{\max} = U_{\max} \cdot \sup_k \bar{A}_k \tag{14}$$

**Assumption 2** (Scale control).

    *a. For all $k$, the scaling parameters satisfy:*

$$0 < \eta_k \leq 1, \quad \xi_k \leq \delta_t \eta_k \quad \text{for some } \delta_t \in [0, 1) \tag{15}$$

    *b. Asymptotically, the schedule satisfies:*

$$\lim_{k \to \infty} \eta_k = 1, \quad \lim_{k \to \infty} \xi_k = 0 \tag{16}$$

**Assumption 3** (Trust region).

$$KL(\pi_k, \pi_{k+1}) \leq \frac{(1 - \gamma)\, \varepsilon_k^2}{2}$$

*(implied by PPO clip ratio $r_\pi \in [1 - \varepsilon_k, 1 + \varepsilon_k]$).*

## C.2 STABILITY AND SAFETY RESULTS

**Lemma 1** (Bounded Shaped Policy Updates). *Under Assumptions 1 (Boundedness), 2 (Scale Control), and 3 (Trust Region), the magnitude of each shaped PPO policy update is uniformly bounded.*

*Proof.* By definition, the shaped advantage satisfies

$$|\tilde{A}_t| \leq \eta_t |A_t| + \xi_t |U_t|.$$

Using Assumption 1 and Assumption 2(a), we obtain

$$|\tilde{A}_t| \leq \eta_t A_{\max} + \xi_t U_{\max} \leq \eta_t A_{\max} + \delta_t \eta_t U_{\max} \leq (1 + \delta_t) A_{\max}.$$

Under the PPO trust-region constraint in Assumption 3, the likelihood ratio is clipped and the score function $\nabla_\theta \log \pi_\theta(a_t|s_t)$ has bounded second moment. Consequently, the policy gradient update

$$\mathcal{L}_k^{shaped} = \mathbb{E}\Big[\nabla_\theta \log \pi_\theta(a_t|s_t)\, \tilde{A}_t\Big]$$

is uniformly bounded in norm. $\square$

**Theorem 2** (Non-Divergence under Trust Region). *Under Assumptions 1 (Boundedness), 2 (Scale Control), and 3 (Trust Region), the PPO updates computed using the shaped advantage $\tilde{A}_t = \eta_t A_t + \xi_t U_t$ remain within the prescribed trust region. In particular, the KL divergence between successive policies is uniformly bounded, and the optimization does not diverge.*

*Proof.* At iteration $k$, PPO computes the policy update by maximizing a clipped surrogate objective of the form

$$\mathcal{L}_k(\theta) = \mathbb{E}\Big[\min\big(r_t(\theta)\, \tilde{A}_t,\ \text{clip}(r_t(\theta), 1 - \varepsilon_k, 1 + \varepsilon_k)\, \tilde{A}_t\big)\Big],$$

where $r_t(\theta) = \pi_\theta(a_t|s_t)/\pi_{\theta_k}(a_t|s_t)$. The clipping operation enforces

$$r_t(\theta) \in [1 - \varepsilon_k,\ 1 + \varepsilon_k],$$

which implies the trust-region bound in Assumption 3 (see Schulman et al. (2017b)).

This constraint depends only on the policy parameterization and the clipping threshold $\varepsilon_k$, and is independent of the specific form of the advantage estimator. Therefore, optimizing the shaped surrogate induces the same likelihood-ratio constraint as standard PPO.

By Lemma 1 (Bounded Shaped Policy Updates), the resulting policy update has uniformly bounded magnitude. Combining bounded update magnitude with the likelihood-ratio constraint yields

$$\text{KL}(\pi_k \,\|\, \pi_{k+1}) \leq \tfrac{(1-\gamma)\varepsilon_k^2}{2}.$$

Hence, the sequence of shaped PPO updates remains within the prescribed trust region, and the optimization does not diverge. $\square$

## C.3 ASYMPTOTIC BEHAVIOR

**Theorem 3** (Asymptotic Equivalence to PPO). *Let* $\{\pi_k\}$ *be the sequence of policies produced by PPO using the shaped advantage*

$$\tilde{A}_t = \eta_t A_t + \xi_t U_t, \tag{17}$$

*where* $U_t$ *is computed on-policy from the same rollouts as* $A_t$. *Under Assumptions 1 (Boundedness), 2 (Scale Control), and 3 (Trust Region), the shaped update is asymptotically equivalent to the standard PPO update. In particular, any stationary point of PPO is also a stationary point of the shaped objective.*

*Proof.* The PPO policy gradient update with shaped advantage is given by

$$\mathcal{L}_k^{shaped} = \mathbb{E}_{(s_t,a_t)\sim\pi_k}\left[\nabla_\theta \log \pi_\theta(a_t|s_t)\,\tilde{A}_t\right]. \tag{18}$$

Substituting the definition of $\tilde{A}_t$ yields

$$\mathcal{L}_k^{shaped} = \eta_k\,\mathcal{L}_k^{\text{PPO}} + \xi_k\,\mathcal{L}_k^U, \tag{19}$$

where

$$\mathcal{L}_k^{\text{PPO}} \doteq \mathbb{E}[\nabla_\theta \log \pi_\theta(a_t|s_t)\,A_t], \tag{20}$$

$$\mathcal{L}_k^U \doteq \mathbb{E}[\nabla_\theta \log \pi_\theta(a_t|s_t)\,U_t]. \tag{21}$$

By Assumption 1, both $A_t$ and $U_t$ are uniformly bounded. Under the trust-region constraint in Assumption 3, the score function $\nabla_\theta \log \pi_\theta(a_t|s_t)$ has bounded second moment, ensuring that both gradient components are finite.

By Assumption 2(b),

$$\lim_{k\to\infty} \eta_k = 1, \qquad \lim_{k\to\infty} \xi_k = 0, \tag{22}$$

which implies

$$\lim_{k\to\infty} \mathcal{L}_k = \mathcal{L}_k^{\text{PPO}}. \tag{23}$$

Therefore, the shaped update converges to the standard PPO update. Any policy $\pi^\star$ satisfying $\mathcal{L}^{\text{PPO}}(\pi^\star) = 0$ is also a stationary point of the shaped objective asymptotically. $\square$

## C.4 EARLY-TRAINING ADVANTAGE

**Theorem**[reinstate] (Non-Vanishing Updates in Sparse-Reward Regimes).

*Proof.* Recall that the shaped advantage is defined as

$$\tilde{A}_t = \eta_k A_t + \xi_k U_t,$$

and the corresponding shaped policy gradient is

$$\mathcal{L}_k^{\text{shaped}} \doteq \mathbb{E}\left[\nabla_\theta \log \pi_\theta(a_t|s_t)\,\tilde{A}_t\right].$$

By linearity of expectation, the shaped gradient decomposes as

$$\mathcal{L}_k^{\text{shaped}} = \eta_k \mathcal{L}_k^{\text{PPO}} + \xi_k \mathcal{L}_k^U, \tag{24}$$

where

$$\mathcal{L}_k^{\text{PPO}} \doteq \mathbb{E}[\nabla_\theta \log \pi_\theta(a_t|s_t)\,A_t], \qquad \mathcal{L}_k^U \doteq \mathbb{E}[\nabla_\theta \log \pi_\theta(a_t|s_t)\,U_t].$$

By assumption, the expected magnitude of the PPO advantage is small, $\mathbb{E}[|A_t|] \leq \varepsilon_A$. Under Assumption 1, $|A_t| \leq A_{\max}$, and under the trust-region constraint (Assumption 3), the score function $\nabla_\theta \log \pi_\theta(a_t|s_t)$ has bounded second moment. Consequently, there exists a constant $C > 0$ such that

$$\left\|\mathcal{L}_k^{\text{PPO}}\right\| \leq C\,\mathbb{E}[|A_t|] \leq C\,\varepsilon_A.$$

FrozenLake involves crossing a frozen lake from the start tile (S) to the goal tile (G) without falling into holes (H), by walking only on frozen tiles (F). The agent begins at state 0, and the goal is located at the last state of the grid. Any move that would take the agent off the board results in staying in place, which is safe and sometimes preferable to risky moves. Given the full grid layout and state representations, provide a trajectory from the start to the goal that avoids all holes and reaches the goal along the shortest safe path. Assume deterministic transitions. Return the best and second-best safe trajectories from start (S) to goal (G) in the form of Python policy dictionaries. Each key should be a state, and each value should be the action to take from that state (0=left, 1=down, 2=right, 3=up). Use None for goal or holes.

```
…. One safe shortest path avoiding all holes is: [0 → 1 → 2 → 6 → 10 → 14 → 15]
Another safe path that's still short (but one step longer): [0 → 1 → 2 → 6 → 10 → 9 → 13 → 14 → 15]
best = { 0: 2, 1: 2, 2: 1, 6: 1, 10: 1, 14: 2, 15: None}
second_best =  {0: 2,1: 2, 2: 1, 6: 1, 10: 0, 9: 1, 13: 2, 14: 2, 15: None}
```
chatGPT o4 mini

Figure 9: Prompt and response from GPT-4o-mini for the FROZENLAKE environment. The LLM receives a serialized map or a textual description and is asked to provide the best and second-best safe trajectories from start to goal, avoiding all holes under deterministic dynamics.

Taking norms on both sides of Eq. equation 24 and applying the triangle inequality yields

$$\left\|\mathcal{L}_k^{\text{shaped}}\right\| \geq \xi_k \left\|\mathcal{L}_k^U\right\| - \eta_k \left\|\mathcal{L}_k^{\text{PPO}}\right\|.$$

Substituting the above bound on $\left\|\mathcal{L}_k^{\text{PPO}}\right\|$ gives

$$\left\|\mathcal{L}_k^{\text{shaped}}\right\| \geq \xi_k \left\|\mathcal{L}_k^U\right\| - \eta_k C \varepsilon_A.$$

Since $0 < \eta_k \leq 1$ by Assumption 2, the second term is $O(\varepsilon_A)$, and we obtain

$$\left\|\mathcal{L}_k^{\text{shaped}}\right\| \geq \xi_k \left\|\mathcal{L}_k^U\right\| - O(\varepsilon_A),$$

which completes the proof. □

**Remark (Bias–Variance Trade-off).** The shaping mechanism can also be interpreted through the lens of the bias–variance trade-off in policy gradient estimation. Standard PPO relies on advantage estimates (e.g., GAE or Monte Carlo returns) that depend on long-horizon rollouts and typically exhibit high variance, particularly in early training. In contrast, the utility signal $U_t$ depends only on the current state–action pair and its similarity to stored high-return trajectories, yielding a lower-variance but biased learning signal. Initializing $\xi_0 > 0$ stabilizes early optimization by injecting this bias, while the decay schedule $\xi_k \to 0$ gradually restores reliance on reward-based advantage estimates as the policy and critic become more reliable.

## D  LLM PROMPTING AND REASONING

### D.1  GYMNASIUM TOY TEXT

FROZENLAKE is a tabular RL environment where the agent starts in the top-left and must reach the bottom-right goal while avoiding holes. For FROZENLAKE, we provide the LLM with the complete map of the environment, either as an image (Figure 8) or as a serialized array representation such as ['F', 'F', ..., 'H', 'F', ..., 'G']. Though the environment is typically stochastic due to slipperiness, the LLM is instructed to assume deterministic transitions.

Although much of the prompt is directly from the official environment description, for clarity and reproducibility, we include the full version. The prompt and the LLM's response are shown in Figure 9.

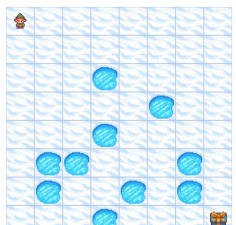

Figure 8: Frozen Lake (Gymnasium)

### D.2  STANDARD AND CUSTOM MINIGRID AND BABYAI ENVIRONMENTS

Each environment was chosen for a specific purpose: REDBALL involves short-horizon navigation and fast spatial goal acquisition. LAVACROSSING introduces irreversible transitions that require long-horizon planning to avoid dead ends. DOORKEY requires the agent to acquire a key, unlock a

> We are in a MiniGrid environment where the agent must pick up a yellow key, use it to toggle a yellow door, and then reach the green goal tile. The action space follows the standard MiniGrid specification.
> This environment is challenging for classical RL algorithms due to its sparse rewards.
> There is only one key-door pair (yellow), and other objects like the purple ball and blue box are distractors.
> Do you understand this environment?
> Answer yes or no.

Figure 10: Prompt to Offline LLM for the custom MiniGrid variant DISTRACTED DOORKEY. The prompt describes the task setting, object roles, and challenges, and asks the LLM to confirm understanding before suggestions.

door, and reach the goal, forming a delayed dependency chain that challenges temporal credit assignment. REDBLUEDOOR tests the agent's ability to commit to a correct action sequence, as opening the blue door prematurely ends the episode. At last, DISTRACTED DOORKEY introduces BabyAI-style distractors (e.g., irrelevant balls and boxes) alongside the original multi-step dependencies of DOORKEY, allowing us to test whether the LLM can generalize across known task elements and maintain coherent subgoal proposals under added visual distraction. For standard MiniGrid and BabyAI environments, we used the environment descriptions provided on the MiniGrid website. For our custom environment (DISTRACTED DOORKEY), we mimicked the phrasing and structure of the official MiniGrid descriptions (Figure 10). Unlike in FROZEN LAKE, obtaining useful trajectories here was not as straightforward. MiniGrid-style environments often required multi-round prompting to obtain meaningful and desired outputs. Moreover, instead of providing an image of the environment, we found it more effective to use a textual description. This helped reduce confusion and encouraged the LLM to understand that object locations (e.g., the key, door, and agent in DOORKEY) can vary across episodes.

## D.3 LLM REASONING PATTERNS ACROSS MODELS

We observed that different LLMs produced very different memory graphs. To better understand how different models reason about these environments, we recorded not only their output trajectories but also their internal reasoning processes. For model that include system-level thinking (e.g., GPT-o4-mini), this was extracted directly from the response. For models that do not expose intermediate reasoning (e.g., Claude 3), we followed up with an auxiliary prompt such as: "Give me your reasoning as to why you chose this sequence of actions."

These responses were not used in the MIRA framework, but we found them surprisingly revealing. Despite receiving identical prompts, the models relied on starkly different reasoning strategies. This divergence gave us unexpected insight into how various LLMs process spatial structure, interpret decision sequences, and reason about reinforcement learning dynamics and learning objectives. Differences that, in turn, shape the quality of their output trajectories. In Figure 11, we present reasoning snippets from the LLMs' outputs. We omit the initial sections where models repeat the prompt or restate the environment description, and instead highlight the specific reasoning steps that led each model to select a particular trajectory. The influence of these differing reasoning strategies on RL performance is reflected in the return curves shown in Figure 7.

## D.4 CASE STUDY: **DISTRACTED DOORKEY**

In the ablation study presented in Subsections 4.2, GPT-o4-mini and Gemini return different outputs when presented with the same situation. Here, we provide the exact prompt and reasoning traces. As shown in Figure 12, both responses appear plausible at a surface level, but only one is consistent with the task dynamics: given that sufficient exploration has already occurred, the key is likely collected, making suppression of the corresponding action the correct response. In this case, the divergence leads to a drop in performance under the misaligned output.

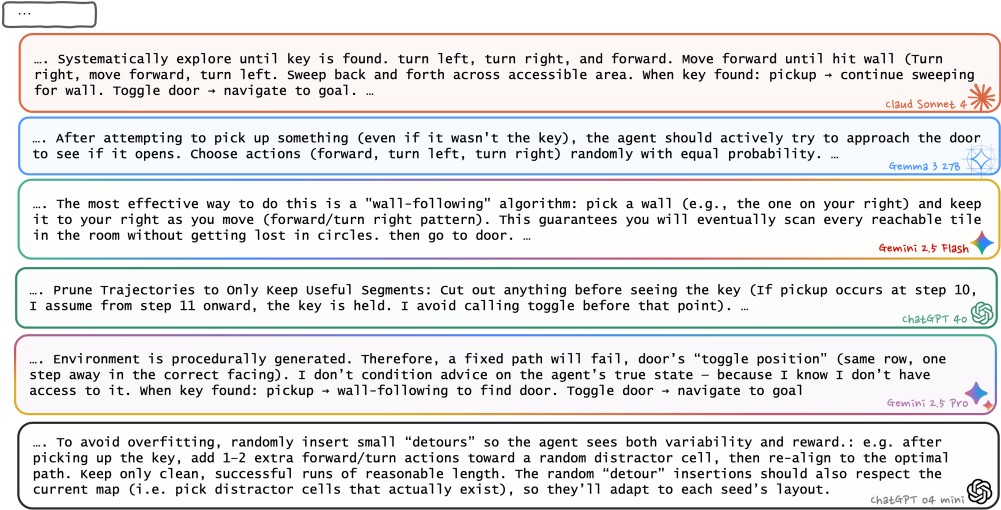

Figure 11: Reasoning traces produced by different LLMs in response to our custom environment prompt as part of "Offline LLM" prompting. After confirming they understood the environment, each model was asked: "If you were to give an RL agent useful trajectories to help solve this, what would you do?" For models that do not output internal reasoning (e.g., Claude), we issued a follow-up prompt requesting their thought process. We omit repeated environment restatements and show only the key parts where the model explains how it decided on the action sequence.

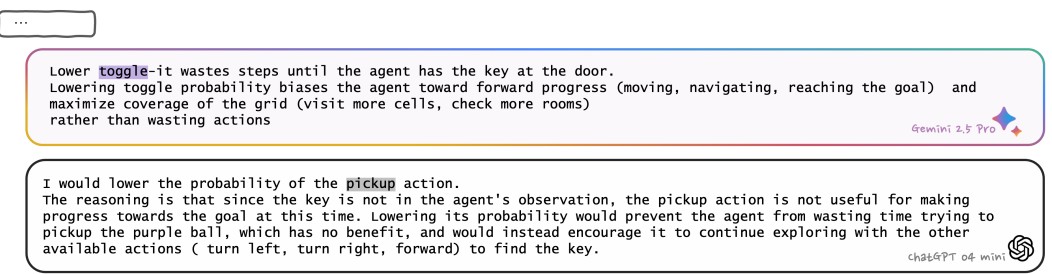

Figure 12: Reasoning traces produced by Gemini and ChatGPT under "Online LLM" prompting. The prompt emphasizes that sufficient exploration has already been performed and, from the partial observation, no key is visible. A (flawed but plausible) line of reasoning is that the agent must still be in the phase of searching for the key, so reducing the probability of toggle appears reasonable to prioritize movement actions for exploration.

## E  MEMORY GRAPH CONSTRUCTION DETAILS

In this section, we further explain the procedure for initializing, updating, and pruning MIRA's memory graph. As discussed in Section 2, the initial memory graph is constructed from offline LLM-generated suggestions. Once built for a specific environment, this graph can be reused across training episodes or even across agents within the same task. Since MIRA is designed to generalize across diverse settings, figure 2 illustrates how the framework accommodates environments with a single terminal objective as well as tasks with multiple independent objectives

Given that each task differs slightly, we largely focus our detailed explanation on DOORKEY from the MiniGrid suite for the rest of the subsections, as it contains multiple subgoals and is sufficiently complex to show the dynamics of the graph clearly.

### E.1 INITIALIZATION

As shown in figure 11, GPT-o4-mini tends to generate trajectory segments that begin after the key is picked up, with the subgoal "toggle the door". In contrast, models like Claude tend to produce longer, full trajectories from the beginning. Interestingly, segmented trajectories are often more useful in this environment. Since the environment is partially observable and reinforcement learning relies heavily on exploration, allowing the agent to figure out how to reach the key on its own helps it understand the overall layout of the environment better. Once the key is acquired, there is a higher chance that the door has already entered the agent's observation window, making memory-guided navigation toward the door more effective.

In addition to segments, the LLM also infers subgoals ($\kappa_\ell$). While the obvious ones are "Pick up key," "Open door," and "Reach goal," o4-mini returns more detailed versions like:

$$\kappa_1 : \textit{Go to key} \rightarrow \kappa_2 : \textit{Pick up key} \rightarrow \kappa_3 : \textit{Go to door} \rightarrow \kappa_4 : \textit{Toggle door} \rightarrow g_\triangleright: \textit{Go to goal.}$$

This fine-grained subgoal sequence reflects the environment's control logic: the "open door" action is valid only if the agent is positioned one step away, properly aligned, and facing the door.

Moreover, for each memory segment, an estimated subgoal reward $\hat{r}_m$ is stored in the node that reflects geometric progress toward completing its associated subgoal. In discrete environments (e.g., MiniGrid), progress is computed using the normalized shortest-path distance between the states in the segment and the subgoal's target location.

### E.2 AGENT-INDUCED UPDATES

During training, new nodes are added to the memory graph or existing ones are updated whenever the agent produces trajectory segments that improve upon what the graph already stores, either by providing a new segment for a (sub)goal or by achieving a higher estimated return than the current entry for that (sub)goal. For example, if the agent independently discovers a shorter path to the key and then follows a memory-guided trajectory to reach the door or goal, the resulting sequence is added as a new node in the graph. Likewise, if the agent successfully executes a trajectory that was initially stored with low confidence from the offline LLM, we treat this behavior as implicit validation and increase the confidence of the corresponding memory node.

The memory graph remains lightweight throughout training. Each node stores a trajectory segment and metadata, and the total graph size stays compact. Compared to experience replay buffers in standard off-policy RL methods, which retain large volumes of data, the memory graph introduces negligible computational and memory overhead. To maintain compactness, unused nodes are periodically pruned based on access frequency. Each memory node tracks an access counter, which is reset every time the node is used. Nodes that are not accessed for 100 episodes are pruned, except for those corresponding to final goal trajectories ($g_\triangleright$), which are retained since the agent might not have reached them early on, but they are essential for guiding successful completion later in training.

Algorithm 2 summarizes the add, update, and prune operations that govern how the memory graph is maintained during training.

### E.3 ONLINE GRAFTING AND TRIGGERS

Since the agent has a limited number of steps per episode, it may fail to reach any subgoal (e.g. "Open Door") with a matching trajectory in the memory graph early on, preventing utility shaping from activating. To address this, MIRA includes a fallback mechanism: if the computed utility $U$ is entirely zero for $N$ consecutive episodes, the agent triggers an online LLM query. These online queries return short plans (e.g., "turn left, move forward, toggle") based on the agent's partial observations to help the agent reorient. Once screened for quality, the new suggestion is grafted into MIRA. Another way online LLM queries contribute is by influencing the agent's policy preferences directly through soft logit injection. Importantly, the online LLM is constrained by the same partial observability as the agent. It does not receive access to the full environment state and therefore cannot, for example, determine the presence of a key elsewhere in the grid. Furthermore, since inventory status is not part of the agent's observation space, the LLM is unaware of whether the

agent has picked up the key. Instead, the LLM receives a batch of recent partial observations and must infer from them whether any meaningful guidance can be offered.

# F  UTILITY COMPUTATION

In this section, we provide a detailed explanation of the utility computation introduced in the main text, clarifying how each component contributes to the shaping term. The utility measures how closely the agent's trajectory aligns with high-return segments stored in the memory graph. When a reference trajectory is matched, utility values are assigned based on reverse-aligned similarity with reference trajectories; unmatched steps receive zero utility. Below, we describe the role of each factor in the computation (Equation 2) and provide pseudocode for the full procedure.

## F.1  SIMILARITY SCORE

The similarity function assigns a score based on the information extracted from the agent's and the reference transition's observations. Depending on the environment, these observations may include position, orientation, or action. For example, in FROZENLAKE, observations are discrete and include only position, so direction is omitted in the similarity check. High similarity indicates that the agent is reproducing a locally meaningful portion of a successful stored trajectory, whereas low or zero similarity reflects small or no meaningful match.

---

**Algorithm 2** Evolving Memory Graph During Training

---

**Require:** Memory graph $\mathcal{G}$, new trajectory and metadata $(\tau', \zeta', \hat{r}', c')$, prune window $W$

$\quad \mathcal{M}_\zeta \leftarrow \{m \in \mathcal{G} : \zeta_m = \zeta'\}$
$\quad$**if** source $=$ ONLINELLM **then**
$\quad\quad$**if** $\neg$SCREENING **then**
$\quad\quad\quad$return $\mathcal{G}$ $\quad \triangleright$ Discard online LLM suggestion
$\quad\quad$**end if**
$\quad$**end if**
$\quad$**if** $\mathcal{M}_\zeta = \emptyset$ $\quad \triangleright$ No existing segment for this subgoal **then**
$\quad\quad$Create node $m' \leftarrow (\tau', \zeta', \hat{r})_{c'}$
$\quad\quad$Initialize access$_{m'} \leftarrow 0$
$\quad\quad$Insert $m'$ into $\mathcal{G}$
$\quad$**else**
$\quad\quad m \leftarrow \arg\max_{m \in \mathcal{M}_\zeta} \hat{r}_m$
$\quad\quad$**if** $\hat{r}' > \hat{r}_m$ **then**
$\quad\quad\quad (\tau_m, \hat{r}_m) \leftarrow (\tau', \hat{r}')$
$\quad\quad\quad c_m \leftarrow c'$
$\quad\quad$**end if**
$\quad\quad$**if** source $=$ AGENT **then**
$\quad\quad\quad c_m \leftarrow \min(1, c_m + \Delta_c)$ $\quad \triangleright$ Agent validation increases confidence
$\quad\quad$**end if**
$\quad$**end if**
$\quad$**for** each node $m \in \mathcal{G}$ **do**
$\quad\quad$**if** access$_m = 0$ for $W$ episodes **then**
$\quad\quad\quad$Remove $m$ from $\mathcal{G}$ $\quad \triangleright$ Prune nodes unused within inactivity window
$\quad\quad$**end if**
$\quad$**end for**
$\quad$return $\mathcal{G}$

---

## F.2  GOAL ALIGNMENT

The goal-alignment factor $\rho$ scales utility based on how closely the subgoal associated with a matched memory segment relates to the subgoal that the current transition corresponds to. Although the RL agent never observes subgoal labels, the *environment state* uniquely identifies which subgoal

---

**Algorithm 3** Similarity Score $\int$

---

**Require:** Agent $x_a$ and Reference $x_m$ annotated transition (metadata)

  **if** (pos., dir.) $\in (o_a, o_m)$ match & $a_a = a_m$ **then**
    **return** $\int = $ high_sim    $\triangleright$ (1)
  **else if** pos. $\in (o_a, o_m)$ match & $a_a = a_m$ **then**
    **return** $\int = $ mod_sim    $\triangleright$ not align direction (0.7)
  **else if** $(d_a \in o_a) \pm 1 \bmod 4 = d_m \in o_m$ **then**
    **return** $\int = $ low_sim    $\triangleright$ action aligned direction (0.4)
  **else**
    **return** $\int = $ no_sim    $\triangleright$ (0)
  **end if**

---

phase the transition belongs to, for example, in DOORKEY environment, whether the agent is still approaching the key or has already picked it up and is proceeding toward the door. Each memory node carries a subgoal label generated by the LLM. These subgoal descriptions consistently identify (i) the object or region involved and (ii) the high-level action applied to it. We extract these two components through simple rule-based parsing over the LLM-generated subgoals, yielding an entity token (e.g., key, door, ball) and an action-phase tag (e.g., navigate). The alignment score $\rho$ is then calculated as the Jaccard similarity between the token pair of the memory node's subgoal and the token pair associated with the transition under evaluation. As a result, locally similar transitions only contribute to utility when they align semantically with the relevant subgoal, preventing behaviorally similar but semantically unrelated memory segments from influencing the utility signal.

---

**Algorithm 4** Goal Alignment $\rho$

---

**Require:** Agent $\zeta_a$ and Reference $\zeta_m$ subgoal

  $t_a \leftarrow $ TOKENS$(\zeta_a)$    $\triangleright$ Agent entity–phase token
  $t_m \leftarrow $ TOKENS$(\zeta_m)$    $\triangleright$ Memory entity–phase token
  $I_\cap \leftarrow t_m \cap t_a$
  $I_\cup \leftarrow t_m \cup t_a$
  **return** $\rho = |I_\cap|/|I_\cup|$

---

With the similarity score $\int$ and alignment factor $\rho$ established, we combine them with the memory-stored quantities $\hat{r}_m$ (estimated subgoal reward) and $c_m$ (LLM confidence) to construct the utility used in advantage shaping.

---

**Algorithm 5** Compute Utility Score

---

**Require:** Agent $\tau_{\text{agent}}$ and Reference $\tau_{\text{m}}$ trajectory

  $x \doteq (o, a, r, \text{meta})$    $\triangleright$ Denote a transition with metadata (e.g. subgoals)
  Initialize $U \leftarrow [0, \ldots, 0]$
  Align the tail of $\tau_{\text{agent}}$ to length of $\tau_{\text{m}}$
  **for** each $(x_a, x_m) \in (\tau_{\text{agent}}^{\text{tail}}, \tau_{\text{m}})$ **do**
    $\int \leftarrow \int((o_a, a_a), (o_m, a_m))$    $\triangleright$ Compute similarity
    $\rho \leftarrow \rho(\zeta_a, \zeta_m)$    $\triangleright$ Compute goal aligment factor
    $u \leftarrow c_m \cdot \hat{r}_m \cdot \rho \cdot \int$
    Assign $u$ to corresponding index in $U$
  **end for**
  **return** $U$

---

# G EXTENDED EXPERIMENTAL STUDIES

## G.1 SENSITIVITY AND ROBUSTNESS STUDIES

### G.1.1 PROMPT ROBUSTNESS

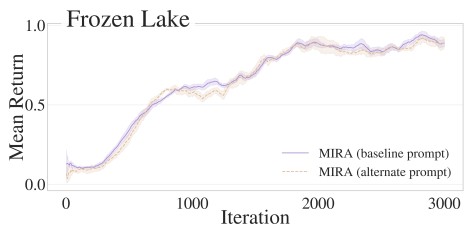

To evaluate robustness to reasonable variations in prompt wording, we repeated the FROZENLAKE experiment using an alternative prompt with simpler phrasing but identical task information. In this variant, the prompt stated that "*FrozenLake is a grid where S is the start, G is the goal, F are safe tiles, and H are holes. The agent moves from state 0 using actions 0=left, 1=down, 2=right, 3=up, and moves that go off the grid keep the agent in place and are safe. Using the grid and assuming deterministic transitions, provide a shortest safe path from S to G that avoids all holes, and return the best and second-best safe paths as Python dictionaries mapping each visited state to its action, using None for the goal or holes.*" Figure 13 compares MIRA under the original and alternative prompts. The learning curves and final returns are closely aligned, showing that MIRA's performance is stable under natural variations in how the environment description is presented.

Figure 13: FrozenLake robustness to prompt wording. MIRA achieves similar performance under the original and alternative prompts, showing stability to natural variations in task description.

### G.1.2 THRESHOLD SENSITIVITY

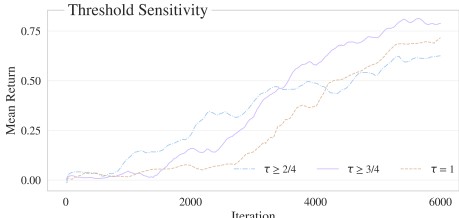

To assess the sensitivity of the screening rule, we conduct a study that varies the acceptance threshold while keeping the overall method fixed. The threshold, therefore, controls when and how densely the graph is populated, not whether shaping exists at all. The main results of the paper uses $k = 3$ LLM completions, which was chosen for efficiency across all experiments. For this diagnostic test only, we set $k = 4$ to obtain cleaner fractional thresholds corresponding to meaningful agreement levels on the DOORKEY environment. We evaluate three settings: a lenient majority rule ($\tau \geq 2/4$), a stricter majority rule ($\tau \geq 3/4$), and unanimous agreement ($\tau = 1$).

Figure 14 shows that the threshold primarily affects the early phase of learning. With a lenient threshold, more candidate suggestions are grafted as healthy graph nodes and added to the memory. This leads the agent to receive utility bonuses on more states while it is still exploring. As a result, even if some early nodes may be slightly misaligned, the overall trend still pushes the agent toward regions that have higher returns. It is then up to the agent to correct those nodes through experience or increase their confidence if they turn out to be helpful, which explains why the overall improvement can be slower in this setting. In the stricter settings, fewer suggestions pass the screening, so graph growth is delayed and the shaping signal remains sparse. Early learning progresses more slowly, but once these high-confidence nodes enter the graph, they produce larger and more coherent utility bonuses along trajectories that already correlate with high return, which creates the steeper rise visible in the mid-training region. The near-final performance still lies in a narrow band, since the shaping term is bounded and eventually dominated by the learned value function. This behavior is consistent with the design of our utility-based shaping, using the LLM's possibly imperfect but often still useful knowledge to accelerate the initial learning phase while ensuring stable progress as more reliable evidence accumulates.

Figure 14: Sensitivity to the screening threshold. Lenient thresholds graft more candidate nodes into memory early, producing broader shaping during exploration but a slower overall improvement rate. Stricter thresholds delay graph growth yet yield sharper mid-training gains once high-confidence nodes appear. All settings converge to a narrow performance band.

## G.2 Early Advantage Dynamics

Figure 15 provides empirical support for the central intuition behind our shaping formulation. We plot return curves for each $\xi$ group (color), across different $\eta$ values (line style). Early in training, return curves within each $\xi$ group remain tightly clustered, indicating that $A_t$, the critic's estimate, provides little useful signal, regardless of how it is weighted. Divergence points, marked on the figure, denote the first iteration where the return spread across $\eta$ values exceeds a certain threshold, signaling that $A_t$ has begun contributing meaningfully to the shaped advantage $\tilde{A}_t = \eta_t A_t + \xi_t U_t$.

In the absence of shaping ($\xi = 0$, gray lines), this occurs relatively late (iteration 131), whereas with shaping ($\xi > 0$), it happens substantially earlier (iterations 81–113, depending on $\xi$). This shows that the utility term not only supports early learning but also accelerates the emergence of a reliable critic. These results validate our choice to softly shape advantages, and emphasize the importance of carefully tuning $\xi$ and $\eta$: insufficient shaping slows critic learning, which in turn leads to substantially lower mean returns.

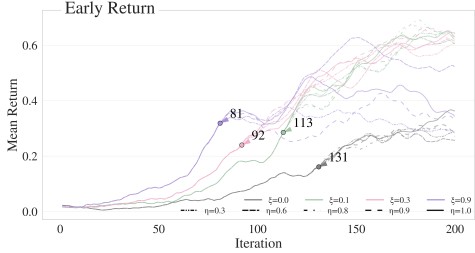

Figure 15: Return curves for different $\eta$ values under fixed $\xi$ settings. Markers indicate the first iteration where performance begins to diverge, signaling when $A_t$ starts to meaningfully affect learning. Early on, the critic signal is weak, and $\tilde{A}_t$ is driven mostly by the utility term. When $\xi$ is large enough, shaping accelerates the critic's contribution by up to 50 iterations and leads to around 2.5× higher return compared to the unshaped case.

**Remark (Optimization Landscape in Sparse-Reward Regimes).** In sparse-reward environments, standard policy gradient methods such as PPO may exhibit near-zero expected gradients in early training, as reward-based advantage estimates $A_t$ are often uninformative until a successful trajectory is observed. The non-vanishing update result (Theorem 1) implies that the proposed shaping objective induces a consistent gradient signal $\xi_k \nabla_\theta \mathbb{E}[U]$ even when reward-based advantages are weak. This additional structure modifies the local optimization landscape by providing an informative descent direction derived from trajectory similarity, thereby facilitating earlier and more stable optimization.

## G.3 Relative Wall Time

We measure relative wall-clock time as the end-to-end runtime per iteration to assess each method's computational burden. Environments with a more complex step logic, such as DISTRACTED DOORKEY, which involves door toggling, key collection, and distractor dynamics, incur higher per-step simulation costs. Tasks like REDBLUEDOOR and LAVACROSSING further increase runtime through frequent failures that trigger repeated episode resets and buffer re-initializations. In contrast, FROZEN LAKE's tabular, low-dimensional transitions execute very quickly, so all methods complete rapidly (we do not run the online variant here since the offline approach suffices). Occasional LLM queries introduce network latency that further raises wall time in the slower domains. As a result, relative wall time grows with both the intrinsic simulation complexity of the environment and any additional algorithmic overhead (e.g., LLM calls).

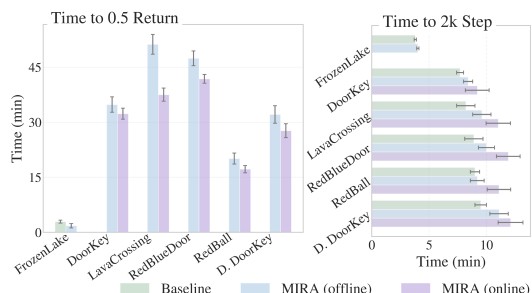

Figure 16: Wall-clock runtimes across environments. Time required to reach a 0.5 return (left): PPO reaches 0.5 only on FROZEN LAKE, while both MIRA variants converge across tasks. Runtime for 2k training steps (right): Online MIRA incurs extra overhead from initial LLM queries, but this cost reduces wasted exploration and leads to faster convergence in terms of overall wall time.

Figure 16 reports wall-clock times for two measures: reaching a 0.5 return (left) and completing a 2k-step run (right). In the left panel, PPO reaches 0.5 only on FROZEN LAKE, while both MIRA variants converge across all environments. In the right panel, PPO shows the lowest per-step runtime because online MIRA incurs some additional cost from its initial LLM queries. However, early queries reduce wasted exploration, allowing online MIRA to reach 0.5 return much faster overall, yielding a net gain in efficiency despite the upfront overhead.

A complete view of these trade-offs comes from considering both Figure 16 and the right-hand panel of Figure 7. Together, they show how higher return and wall-clock time interact when online LLM latency is present. Although online MIRA incurs additional latency from occasional queries, a substantial portion of this cost is offset by faster policy improvement: the agent spends less time in unproductive exploration and reaches competent behavior sooner. This can be seen directly by comparing the left and right panels of Figure 16: online MIRA has slightly higher per-step runtime, yet it reaches the 0.5 return threshold earlier in wall-clock time. The right-hand panel of Figure 7 reinforces this result, showing that the higher cost of online queries is compensated by more rapid performance gains.

## G.4 MEMORY GROWTH ANALYSIS

We examine how the memory graph evolves during training. MIRA updates the graph only when computing the utility term during advantage estimation, so memory expansion occurs once per training batch rather than at every environment step. Because batches contain full trajectories, their lengths vary across environments. To obtain a consistent summary across tasks, we record the memory size every 100 training iterations.

Figure 17 reports memory growth for REDBALL, DOORKEY, and DISTRACTED DOORKEY. In all three environments, memory grows quickly during early training, reflecting the period where the agent encounters diverse high-return segments and, when

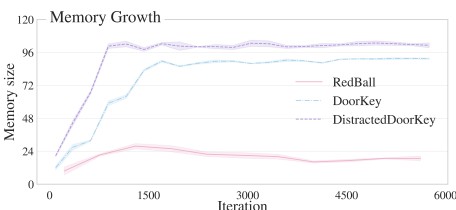

Figure 17: Memory size growth over training. Early growth is followed by convergence once trajectories become consistent, with final sizes increasing from REDBALL to DOORKEY to DISTRACTED DOORKEY in line with task complexity.

applicable, issues LLM queries that generate candidate memory additions. As trajectories converge to a consistent solution, memory expansion slows and eventually stabilizes. The plateau levels reflect the structural demands of each task: REDBALL retains only a small set of nodes due to its simple subgoal structure, whereas DOORKEY and DISTRACTED DOORKEY require a richer collection of region-anchored segments. DISTRACTED DOORKEY ends up with the largest memory, as the distractor objects create additional path variants that help guide the agent, while the total size remains bounded and does not grow throughout later training.

## G.5 QUERY FREQUENCY PERFORMANCE SUMMARY

Table 1 expands on Figure 7 in Subsection 4.2. It shows how different online query budgets impact learning progress (SR90Return, indicating the mean return when success rate first exceeds 90%), final return, and convergence speed (total steps to termination). The results reinforce that while all MIRA variants outperform PPO, higher online budgets further accelerate training and improve asymptotic performance.

Table 1: Performance on DOORKEY. SR90Return is the mean return when success rate first exceeds 90%; Final Return is the return at the end of training; Final Step is the total environment steps. MIRA variants outperform the baseline in both early and final return, with MIRA (large) achieving the highest values while converging fastest.

| Method | SR90Return↑ | Final Return↑ | Final Step↓ |
|---|---|---|---|
| Baseline | $0 \pm 0.002$ | $0.009 \pm 0.001$ | 10362 |
| MIRA (offline) | $0.233 \pm 0.087$ | $0.295 \pm 0.123$ | 10351 |
| MIRA (mid) | $0.284 \pm 0.065$ | $0.902 \pm 0.012$ | 10257 |
| MIRA (large) | $0.851 \pm 0.060$ | $0.91 \pm 0.013$ | 9961 |

### G.6 MINIGRID PERFORMANCE SUMMARY

Tables 2 and 3 report detailed numerical results for all four MiniGrid tasks, including mean returns and success rates averaged over unseen seeds. MIRA consistently outperforms both PPO and the hierarchical baseline across all environments, including the more complex ones such as DOORKEY and REDBLUEDOOR. Welch's t-tests (Ruxton, 2006) show no statistically significant difference between MIRA and LLM4Teach at the 0.05 level across metrics and environments (Table 4). These results support the aggregate performance trends in the main text (Figure 6), demonstrating that MIRA improves both final return and task completion.

Table 2: Mean return on unseen seeds across MiniGrid environments. MIRA achieves high and stable success, comparable to LLM4Teach, despite requiring substantially fewer LLM queries.

| Method | DOORKEY | LAVACROSSING | REDBLUEDOOR | REDBALL |
|---|---|---|---|---|
| Baseline RL | $0.018 \pm 0.016$ | $0.012 \pm 0.027$ | $0.044 \pm 0.042$ | $0.329 \pm 0.205$ |
| HRL | $0.852 \pm 0.017$ | $0.798 \pm 0.090$ | $0.830 \pm 0.021$ | $0.939 \pm 0.046$ |
| LLM4Teach | $\mathbf{0.912} \pm 0.075$ | $\mathbf{0.884} \pm 0.100$ | $0.901 \pm 0.082$ | $\mathbf{0.946} \pm 0.051$ |
| **MIRA** | $0.898 \pm 0.093$ | $0.855 \pm 0.132$ | $\mathbf{0.911} \pm 0.077$ | $0.942 \pm 0.054$ |

### G.6.1 T-TEST: MIRA VS. LLM4TEACH

To assess whether the performance differences between LLM4Teach and MIRA are statistically significant, we conduct Welch's t-tests on the evaluation metrics across environments and seeds. Welch's t-test is a two-sample statistical test that does not assume equal variance. As shown in Table 4, none of the differences reach significance at the $\alpha = 0.05$ level. This suggests that MIRA performs comparably to LLM4Teach across all reported metrics, despite MIRA having small lower final reward.

## H LIMITATIONS

While MIRA improves sample efficiency and reduces reliance on frequent LLM queries, it also comes with natural trade-offs. The method relies on offline LLM outputs to initialize its memory graph, which, if they include misleading information or are not well aligned with the environment dynamics, can slow convergence or increase the need for online queries. Our screening and pruning mechanisms reduce this risk, and in practice it is largely a limitation of current LLMs that is expected to diminish as models improve. MIRA also introduces shaping terms that require hyperparameter tuning to avoid instability between the actor and critic. We find, however, that they can be adjusted with standard procedures. Finally, our current study focuses on discrete action spaces; extending MIRA to continuous domains without discretization is a natural next step.

## I REPRODUCIBILITY

Experiments were run on both a Linux server with Intel Xeon E5-2630 v4 CPUs (40 threads) and an Apple M2 (8-core CPU, 10-core GPU, 16GB unified memory). All LLM models used in our experiments correspond to the publicly available versions released in the first week of August 2025.

### I.1 SIMULATION PLATFORMS

### I.1.1 GYMNASIUM TOY TEXT

ENVIRONMENT DETAILS. Horizon indicates the maximum number of steps per episode before automatic termination (i.e., maxsteps in the environment configuration).

Table 3: Success rate on unseen seeds across MiniGrid environments. MIRA achieves consistently high success rates, matching LLM4Teach while requiring fewer queries, and outperforming baseline and HRL methods.

| Method | DOORKEY | LAVACROSSING | REDBLUEDOOR | REDBALL |
|---|---|---|---|---|
| Baseline RL | $0.023 \pm 0.017$ | $0.017 \pm 0.020$ | $0.036 \pm 0.043$ | $0.539 \pm 0.064$ |
| HRL | $0.897 \pm 0.013$ | $0.841 \pm 0.085$ | $0.892 \pm 0.012$ | $0.956 \pm 0.025$ |
| LLM4Teach | $0.970 \pm 0.004$ | $0.931 \pm 0.011$ | $0.956 \pm 0.003$ | $0.958 \pm 0.021$ |
| **MIRA** | $0.953 \pm 0.043$ | $0.913 \pm 0.077$ | $0.944 \pm 0.020$ | $0.956 \pm 0.036$ |

Table 4: Welch's t-test comparing LLM4Teach and MIRA (MR: Mean Return - SR: Success Rate). None of the differences are statistically significant at $\alpha = 0.05$.

| Metric | LLM4Teach | MIRA | t | p | 95% CI |
|---|---|---|---|---|---|
| DOORKEY (MR) | $0.912 \pm 0.075$ | $0.898 \pm 0.093$ | 0.203 | 0.8495 | [–0.181, 0.209] |
| DOORKEY (SR) | $0.970 \pm 0.004$ | $0.953 \pm 0.043$ | 0.682 | 0.5647 | [–0.0885, 0.1225] |
| LAVACROSSING (MR) | $0.884 \pm 0.100$ | $0.855 \pm 0.132$ | 0.303 | 0.7778 | [–0.2443, 0.3023] |
| LAVACROSSING (SR) | $0.931 \pm 0.011$ | $0.913 \pm 0.077$ | 0.401 | 0.7260 | [–0.1681, 0.2041] |
| REDBLUEDOOR (MR) | $0.901 \pm 0.082$ | $0.911 \pm 0.077$ | –0.154 | 0.8851 | [–0.1906, 0.1706] |
| REDBLUEDOOR (SR) | $0.956 \pm 0.003$ | $0.944 \pm 0.020$ | 1.028 | 0.4081 | [–0.0362, 0.0602] |
| REDBALL (MR) | $0.946 \pm 0.051$ | $0.942 \pm 0.054$ | 0.093 | 0.9302 | [–0.1152, 0.1232] |
| REDBALL (SR) | $0.958 \pm 0.021$ | $0.956 \pm 0.036$ | 0.083 | 0.9387 | [–0.0717, 0.0757] |

Table 5: FrozenLake environment details.

| Property | Value |
|---|---|
| Observation Type | Discrete |
| Horizon | 200 |
| Reward Sparsity | Sparse |
| Action Space | 4 (tabular) |
| Dynamics | Slippery, irreversible |

HYPERPARAMETER. Table 6 provides the main specifications of FrozenLake for `PPOConfig` in RLlib.

Table 6: Hyperparameters of FROZENLAKE

| Parameter | Value |
|---|---|
| Learning rate | $1 \times 10^{-4}$ |
| Batch size | 512 |
| Mini-batch size | 64 |
| Number of epochs | 4 |
| Entropy coefficient | 0.01 |
| Discount factor ($\gamma$) | 0.99 |
| GAE lambda ($\lambda$) | 0.95 |
| Utility ($\xi$) | [0.9] |
| Batch mode | "complete episodes" |

### I.1.2 MINIGRID AND BABYAI

ENVIRONMENT DETAILS. Horizon indicates the maximum number of steps per episode before automatic termination (i.e., maxsteps in the environment configuration).

Table 9: Hyperparameters of DOORKEY

| Parameter | Value |
|---|---|
| Learning rate | $2.5 \times 10^{-4}$ |
| Batch size | 1024 |
| Mini-batch size | 64 |
| Number of epochs | 4 |
| Entropy coefficient | 0.01 |
| Discount factor ($\gamma$) | 0.99 |
| GAE lambda ($\lambda$) | 0.95 |
| Utility ($\xi$) | [0.25, 0.15] |
| Batch mode | "complete episodes" |

Table 10: Hyperparameters of LAVACROSSING

| Parameter | Value |
|---|---|
| Learning rate | $2.5 \times 10^{-4}$ |
| Batch size | 1024 |
| Mini-batch size | 64 |
| Number of epochs | 4 |
| Entropy coefficient | 0.01 |
| Discount factor ($\gamma$) | 0.99 |
| GAE lambda ($\lambda$) | 0.95 |
| Utility ($\xi$) | [0.3] |
| Batch mode | "complete episodes" |

Table 11: Hyperparameters of REDBLUEDOOR

| Parameter | Value |
|---|---|
| Learning rate | $5 \times 10^{-5}$ |
| Batch size | 1024 |
| Mini-batch size | 64 |
| Number of epochs | 4 |
| Entropy coefficient | 0.01 |
| Discount factor ($\gamma$) | 0.99 |
| GAE lambda ($\lambda$) | 0.9 |
| Utility ($\xi$) | [0.25] |
| Batch mode | "complete episodes" |

Table 12: Hyperparameters of REDBALL

| Parameter | Value |
|---|---|
| Learning rate | $2 \times 10^{-4}$ |
| Batch size | 512 |
| Mini-batch size | 64 |
| Number of epochs | 4 |
| Entropy coefficient | 0.01 |
| Discount factor ($\gamma$) | 0.99 |
| GAE lambda ($\lambda$) | 0.95 |
| Utility ($\xi$) | [0.2] |
| Batch mode | "complete episodes" |

Table 7: MiniGrid suite details.

| Property | Value |
|---|---|
| Observation Type | RGB |
| Reward Sparsity | Sparse and delayed |
| Action Space | 7 (tabular) |
| View Size | 7 |
| Horizon | 300 |

Table 8: MiniGrid environments and their dynamics.

| Environment | Dynamics |
|---|---|
| REDBALL | Reversible |
| REDBLUEDOOR | Irreversible |
| LAVACROSSING | Irreversible |
| DOORKEY | Subgoal seq. |
| DISTRACTED DOORKEY | +Visual distractors |

HYPERPARAMETER. Tables 10- 12 provides the main specifications of all the MiniGrid environments for `PPOConfig` in RLlib.

OBSERVATION SPACE In MiniGrid environments, the agent receives an RGB image of the grid, which is passed through a convolutional encoder 18 to extract spatial features relevant for navigation and interaction.

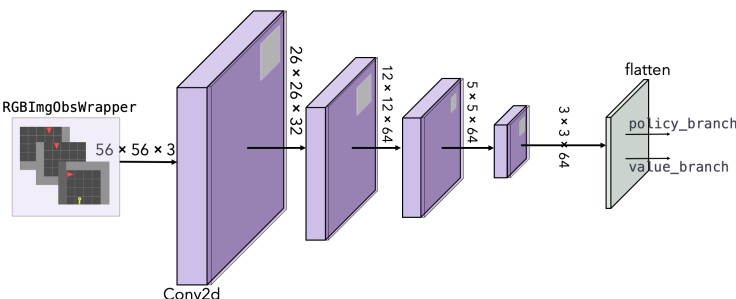

Figure 18: Convolutional encoder architecture used to process the agent's $56 \times 56 \times 3$ RGB observation in MiniGrid environments. The input passes through a series of Conv2D layers, reducing spatial dimensions while increasing channel depth. The final activation is flattened and fed to both policy and value heads. This encoder captures spatial layout, object presence, and agent-centric context for decision-making.

This CNN processes the visual input into a compact feature vector, capturing object positions, colors, and layout structure. The resulting embedding is concatenated with a learned directional encoding and passed to the policy and value heads for action selection and value estimation.

### I.2 LLM CONFIDENCE SETTINGS

For completions where token-level likelihoods are available, confidence is computed using an exponential of the geometric-mean log-probability ($\exp\left(1/L \sum \log p_i\right)$) with a fixed likelihood threshold of $\tau \geq 0.65$. When likelihoods are unavailable, we obtain k independent completions ($k = 3$) and retain only outputs that pass a majority-consistency test with a fixed agreement threshold $\tau \geq 2/3$.

## USE OF LARGE LANGUAGE MODELS (LLMS)

During the preparation of this manuscript, the authors used OpenAI's ChatGPT to assist with grammar and readability. No research ideas, technical content, or analysis were generated by the tool. All content was reviewed and verified by the authors, who take full responsibility for the final version.

