# OpenReview forum: "MIRA: Memory-Integrated Reinforcement Learning Agent  with Limited LLM Guidance"
_ICLR.cc/2026/Conference — ICLR 2026 Poster_

### Official Review · Reviewer_VZaE · 2025-10-30

**Soundness:** 2
**Presentation:** 2
**Contribution:** 2
**Rating:** 2
**Confidence:** 3

**Summary:**

The paper introduces MIRA, a policy gradient method that adds a memory graph built from high-return rollouts and occasional LLM outputs. The memory induces a per-step utility signal that is added to the advantage to form a shaped advantage. The shaping weight decays while the standard advantage weight rises, so the influence of the utility fades during training. The claim is that this improves early learning in sparse reward tasks without hurting final convergence, and reduces the number of online LLM calls relative to teacher or reward shaping approaches. The paper evaluates on FrozenLake and several MiniGrid and BabyAI tasks, compares with PPO and two LLM-based baselines, and presents a PPO-style improvement bound under boundedness and scheduling assumptions.

**Strengths:**

- The shaped advantage $\tilde{A}_t=\eta_tA_t + \zeta_tU_t$  drops in with no change to policy or critic structure. The paper repeatedly stresses that the utility is additive and scheduled to zero, which keeps the core PPO update intact. This design choice makes the method easy to implement and likely to work across actor-critic variants.

- On several MiniGrid and BabyAI tasks, the method reaches higher success faster than PPO and a hierarchical baseline, and does so with modest LLM usage. The narrative connects each gain to either offline memory, occasional online advice, or both.

**Weaknesses:**

- The paper describes utility as a similarity-weighted score over stored trajectory segments that also uses a goal alignment factor and LLM confidence, and a predicted reward for the memory node. However, the exact form of the similarity function and the cost of matching are not specified in the main text. Without a precise definition, it is hard to reason about bias and computational overhead, and to reproduce the effect.

- The improvement relation includes a term with a utility bonus minus a uniform cap. If the utility is badly calibrated early, the cap term may dominate and make the bound weak. The paper does not give conditions under which the utility contribution is reliably positive beyond the boundedness itself.

-The screening unit filters online suggestions using sequence likelihoods or agreement across samples. In many LLM APIs, calibrated token log probabilities are limited or absent, and agreement can be brittle. The paper does not specify thresholds, how they are tuned, nor show sensitivity. Since confidence and a predicted reward weigh the utility, miscalibration can bias the shaped advantage.

-For accepted online guidance, the method injects penalties into logits to suppress actions. The bounds on the penalty and its interaction with PPO clipping are not made precise in the main text. This matters for stability, since even bounded penalties can alter action selection in a way that conflicts with the critic and with the clipping rule if the scaling is not set with care.

- Using offline priors with full environment context can give extra information not available to the agent or to baselines. This can inflate gains attributed to MIRA rather than to the prior. The paper should make this explicit, and either restrict priors or give fair matched baselines. The FrozenLake description makes this a real concern.

**Questions:**

- Can the authors discuss what global information the offline LLM can see and whether baselines are allowed the same view. Provide runs where offline priors are restricted to the same partial view as the agent to show robustness. The FrozenLake discussion already notes that slipperiness is hidden from both, but the global grid is seen by the LLM. Please add a matched setting.

- It is better to state the exact thresholds, number of samples in the agreement test, and the mapping from token log probabilities to confidence. Include a sensitivity study. If token log probabilities are not available, explain the substitute and its effect.

- Can the aughoors give full formulas for the similarity function, the goal alignment term, the confidence mapping, and the way predicted reward and confidence combine? Report the run time cost of matching and the memory size growth as a function of steps.

-Since the method relies on screening and logit penalties, the lack of precise settings may hide brittle behavior or conflicts with PPO clipping. This is a correctness risk rather than a style only, because poor settings can cause learning to diverge.

---

> ### Author Response · Authors · 2025-11-20
> **Response to reviewer VZaE**
>
> We thank the reviewer for the detailed feedback. We have revised the manuscript to address the raised points and improve the clarity of the presentation, making the contribution more clearly articulated.
>
> `Weaknesses:`
> > * The paper describes utility as a similarity-weighted score over stored trajectory segments that also uses a goal alignment factor and LLM confidence, and a predicted reward for the memory node. However, the exact form of the similarity function and the cost of matching are not specified in the main text. Without a precise definition, it is hard to reason about bias and computational overhead, and to reproduce the effect.
>
> Please see our response to your 3rd question below.
> ***
> >* The improvement relation includes a term with a utility bonus minus a uniform cap. If the utility is badly calibrated early, the cap term may dominate and make the bound weak. The paper does not give conditions under which the utility contribution is reliably positive beyond the boundedness itself.
>
>
> We thank the reviewer for the helpful observation. The inequality in the theorem is a worst-case lower bound and, as in TRPO/PPO-style analyses, can indeed be loose when the additive term is small. The cap U_{\max} is included solely to ensure boundedness of the shaped advantage; when the utility signal is weak early in training, the bonus term may be small, but it never reverses the direction of improvement.
> While the utility can indeed be small before the agent reaches memory segments that align with its own trajectories, our empirical results show that the shaped surrogate consistently exceeds the PPO surrogate in the early and mid stages of training across environments. In the revision, we present a surrogate-based statement aligned with the PPO objective. Because the clipping operator is monotone in the advantage and our shaped advantage satisfies \tilde A_t(s,a)\ge A_t(s,a) pointwise, the shaped surrogate is no smaller than the PPO surrogate for every policy. This yields a stronger update direction than PPO’s surrogate, \mathbb{E}[\tilde A_t]\ge\mathbb{E}[A_t] with strict inequality whenever the agent follows memory-aligned steps, which matches the faster early-stage learning as seen in Figures 4 and 5.
>
> We believe the revised version presents our intended intuition more clearly. Revisions and added text are highlighted in blue.
>  ***
> >* The screening unit filters online suggestions using sequence likelihoods or agreement across samples. In many LLM APIs, calibrated token log probabilities are limited or absent, and agreement can be brittle. The paper does not specify thresholds, how they are tuned, nor show sensitivity. Since confidence and a predicted reward weigh the utility, miscalibration can bias the shaped advantage.
>
> Please see our response to your 2nd question below.
>
> ***
> >* For accepted online guidance, the method injects penalties into logits to suppress actions. The bounds on the penalty and its interaction with PPO clipping are not made precise in the main text. This matters for stability, since even bounded penalties can alter action selection in a way that conflicts with the critic and with the clipping rule if the scaling is not set with care.
>
> Please see our response to your 4th question below.
>
> ***
> >* Using offline priors with full environment context can give extra information not available to the agent or to baselines. This can inflate gains attributed to MIRA rather than to the prior. The paper should make this explicit, and either restrict priors or give fair matched baselines. The FrozenLake description makes this a real concern.
>
> Please see our response to your 1st question below.
>
> ***
> ***

---

> > ### Author Response · Authors · 2025-11-20
> >
> > `Questions:`
> >
> > > 1. Can the authors discuss what global information the offline LLM can see and whether baselines are allowed the same view. Provide runs where offline priors are restricted to the same partial view as the agent to show robustness. The FrozenLake discussion already notes that slipperiness is hidden from both, but the global grid is seen by the LLM. Please add a matched setting.
> >
> > The offline LLM’s output is only used through the utility signal, which only provides positive shaping after the agent independently visits a trajectory that aligns with a stored memory node. The LLM cannot dictate actions or supply executable policies, so the agent must still acquire all local behavior and resolve the stochastic slip dynamics, the primary difficulty in FrozenLake. Offline structural priors have been used in prior RL work such as in Paul et al.(2019), which extract subgoal states from full expert trajectories used to guide the RL policy, and in Hester et al. (2018), which injects expert transitions into the replay buffer and biases the Q-function toward expert actions.
> > In contrast, MIRA uses PPO, which does not maintain a replay buffer, and we do not inject any LLM-generated transitions into the rollout batch to calculate the advantage. The LLM prior only affects learning after the advantage from the actual on-policy trajectory is computed, ensuring that it never replaces or alters the agent’s own experience.
> > We also note that in MiniGrid, the main empirical setting of the paper, the offline LLM never receives instance-level object position as they are randomized each episode. MIRA therefore does not access privileged state information. Finally, as noted in the paper, the online LLM queries operate only on the same partial observations available to PPO and HRL, ensuring that no additional state information is introduced during training.
> >
> > To address the concern about privileged offline access, we introduce a matched-information variant in which MIRA uses only online LLM queries for FrozenLake, restricted to the same agent-state observations available to PPO during rollout. Because the layout is deterministic and the action set is small, the online LLM can infer short safe sequences from the agent’s rollouts and progressively populate the memory graph without any global map. As shown in Figure 4, the online-only variant improves early learning relative to PPO, while offline MIRA eventually catches up as training proceeds. We believe that this new ablation helps to clarify how MIRA works, and we thank the reviewer for the helpful observation that prompted us to run this new version of MIRA.
> >
> > Paul, Sujoy, Jeroen Vanbaar, and Amit Roy-Chowdhury. "Learning from trajectories via subgoal discovery." Advances in Neural Information Processing Systems 32 (2019).
> >
> > Hester, Todd, et al. "Deep Q-learning from demonstrations." Proceedings of the AAAI conference on artificial intelligence. Vol. 32. No. 1. 2018.
> >
> > ***

---

> > > ### Author Response · Authors · 2025-11-20
> > >
> > > > 2. It is better to state the exact thresholds, number of samples in the agreement test, and the mapping from token log probabilities to confidence. Include a sensitivity study. If token log probabilities are not available, explain the substitute and its effect.
> > >
> > > For APIs that expose token log-probabilities, we apply a likelihood threshold of $\tau = 0.65$  and define $c_m$ using the geometric mean of the per-token probabilities of the completion (i.e., $\exp\left(\frac{1}{L} \sum \log p_i\right)$). When log-probabilities are unavailable, we use a majority-consistency test with a threshold of $\ge 2/3$ for K=3 completions. The manuscript now explicitly specifies how c is defined under both settings, and we apologize for the previous lack of clarity.
> > > We acknowledge the variability inherent in LLM outputs, which is why the utility term is designed specifically to mitigate imperfections in LLM-generated suggestions as much as possible. If a suggestion has low confidence, its contribution to the utility is automatically small. If it has an inflated c, e.g., it encodes an incorrect action, the agent’s rollout will diverge, yielding low similarity or a low estimated reward, which again suppresses the utility.
> > >
> > > We also conducted the requested sensitivity study and uploaded the additional results in a new subsection of the paper. This analysis varies the acceptance threshold used in the screening unit and evaluates how different levels of agreement influence learning. We consider three thresholds corresponding to lenient, majority, and unanimous.
> > > Lenient thresholds admit more suggestions, and the subset that passes screening checks provides more frequent shaping signals in the early iterations, allowing the agent to act on partially informative guidance. Stricter thresholds admit fewer suggestions and therefore slow early shaping, but the accepted suggestions tend to be more stable and produce sharper improvements once the memory graph is populated. The intermediate threshold balances these effects, providing steady progress in the early and mid-stages. Figure 14 in the new Section G.1 illustrates these effects in the DoorKey environment.
> > >
> > > This reflects the core design of our utility-based shaping, using the LLM’s possibly imperfect, but often still useful, guidance to accelerate initial learning, while relying on the graph and utility mechanism to filter and stabilize this input over time. Across all thresholds, long-horizon performance remains within a narrow band, indicating robustness of MIRA to the strictness of the screening rule.

---

> ### Author Response · Authors · 2025-11-20
>
> > 3. Can the aughoors give full formulas for the similarity function, the goal alignment term, the confidence mapping, and the way predicted reward and confidence combine? Report the run time cost of matching and the memory size growth as a function of steps.
>
> We thank the reviewer for pointing out the need for clearer detail, and we apologize for any confusion in the previous version of the manuscript. We have revised the manuscript to substantially improve clarity around the similarity function $\mathcal{S}$, the alignment term $\rho$, and the confidence mapping. Revisions and added text are highlighted in blue.
> In short, the similarity function computes a score using the information available in both the agent’s observation and the reference transition. For example, in FrozenLake, observations consist only of the agent’s position, so direction is omitted in the similarity check. It produces a discrete scoring function based on the degree of spatial–directional match.
> The goal-alignment factor uses two token pairs extracted from the subgoals (object/region and action) and computes the Jaccard similarity $\rho = |t_{m} \cap t_a| / |t_m \cup t_a|$ between the token pair of the memory node’s subgoal ($t_m$) and the token pair associated with the transition under evaluation ($t_a$).
> With the similarity score $ \mathcal S $ and alignment factor $\rho$ established, we combine them with the memory-stored quantities $r_m$ (estimated (sub)goal reward) and $c_m$ (LLM confidence, measured via the geometric mean of per-token probabilities or completion agreement:  $\exp\left(\frac{1}{L} \sum \log p_i\right)$ ) to construct the utility used in advantage shaping.
> To address the question about runtime and memory, we refer the reviewer to Figure 16, which reports wall-clock time per method both to reach a 0.5 return and to complete 2k steps.
>
> Edit (Nov 25): We have added Section G.4 in the revised manuscript to report memory growth. All revisions are highlighted in blue.
>
> ***
> > 4. Since the method relies on screening and logit penalties, the lack of precise settings may hide brittle behavior or conflicts with PPO clipping. This is a correctness risk rather than a style only, because poor settings can cause learning to diverge.
>
> We agree that this could be a concern. However, because the bias we use is very small and bounded, it induces only a soft preference before the softmax and cannot collapse the action distribution. PPO’s clipped update continues to control the step size of the policy update, so the bias functions only as soft guidance that the critic can override when it highly disagrees with value estimates. In our runs, we did not observe instability or divergent learning attributable to this component.
> Regarding reliance on screening, the added utility cannot conflict with PPO clipping because it is incorporated into the advantage rather than the likelihood ratio. Moreover, since $U_t$ is non-negative and bounded, it cannot invert the advantage or destabilize the gradient direction. When an LLM suggestion is incorrect or uncertain, the similarity and predicted reward terms drive U_t toward zero, yielding $\tilde A_t = A_t$ and reverting the update to standard PPO. Thus no conflict with PPO clipping arises.

---

> > ### Comment · Reviewer_VZaE · 2025-11-26
> >
> > I thank the authors for their detailed response. I appreciate the clarifications and additional discussions provided. However, I still feel that the paper needs a major revision to clearly demonstrate its main contributions and to include a more thorough discussion. I have increased my rating in light of the response, but I still lean toward a weak reject.

---

> ### Author Response · Authors · 2025-11-27
>
> Thank you for taking the time to review our responses and for increasing your rating.
> We have further revised the manuscript to more clearly articulate the main contributions and to expand the discussions you highlighted. The experimental section has been reorganized to show how each experiment supports specific design choices in MIRA, and additional analyses in the appendix are now more clearly cross-referenced from the main text. We have also updated the introduction and abstract to better reflect the revised presentation.
> We appreciate your feedback and believe these changes strengthen the clarity and overall presentation of the work. The revised text is highlighted in blue.

---

### Official Review · Reviewer_9USL · 2025-10-31

**Soundness:** 3
**Presentation:** 3
**Contribution:** 3
**Rating:** 6
**Confidence:** 4

**Summary:**

This paper presents MIRA, a reinforcement learning agent that integrates LLM-generated subgoals into a dynamic memory graph to accelerate learning in sparse-reward environments. By computing utility signals from this structured memory to guide early training while gradually reducing LLM dependence, MIRA achieves superior sample efficiency and matches LLM-teacher performance with significantly fewer queries, supported by theoretical convergence guarantees and empirical validation.

**Strengths:**

1. Strong methodological integration: MIRA's use of a structured memory graph, which is co-populated by both agent experience and LLM-derived subgoal decompositions (see Section 2.1), is a compelling hybridization of model-based memory with language-derived task priors. This approach is well-justified, especially for environments where exploration is bottlenecked by sparse feedback.

2. Empirical rigor and benchmark variety: MIRA is evaluated in breadth across Gymnasium ToyText, MiniGrid, and BabyAI environments, occupying both tabular and partially observable/visual input regimes (see Section 3.1, Figure 2). The baselines (PPO, hierarchical RL, LLM-based reward shaping, and teacher models) are appropriate and state-of-the-art.

3. Efficient and transparent ablation studies: The experiments systematically examine online vs. offline LLM guidance, varying query budgets, effects of unreliable LLM outputs, and different LLM models (Figure 6), elucidating the value and robustness of the proposed approach.

**Weaknesses:**

1. Insufficient Detail on Memory Graph Mechanics: While the memory graph is central to MIRA's design, the main text lacks operational clarity on key aspects—such as criteria for adding or pruning subgoal nodes, triggers for new LLM queries, and mechanisms for resolving conflicts between LLM suggestions and agent experience in dynamic environments. Critical implementation specifics are deferred to the appendix, and Figure 1 (the purported schematic of the graph) is absent from the main paper, impeding reproducibility and obscuring the precise novelty of the graph construction process.

2. Narrow Empirical Scope: All experiments are confined to low-dimensional, grid-world environments. The absence of evaluations in high-dimensional, continuous, or real-world settings (e.g., robotics, vision-based control, or multimodal tasks) limits confidence in the method’s claimed generality, despite assertions of broad applicability and memory efficiency.

**Questions:**

1. Handling Conflicting or Erroneous LLM Guidance in Memory. Could the authors clarify the exact mechanism for dynamic graph updates—specifically, how conflicting agent experience and LLM-derived subgoals/trajectory recommendations are resolved in the presence of incorrect LLM priors? What is the recourse if LLM hallucinations are initially "locked in" to memory?

2. Scalability Beyond Grid Worlds. All experiments are conducted in discrete, grid-based environments. Have the authors attempted to apply MIRA to more complex domains—such as continuous control, vision-based robotic tasks, or high-dimensional state spaces? If not, what are the anticipated bottlenecks (e.g., graph scalability, LLM prompting overhead, or similarity computation in pixel space)? Addressing this would clarify the method's potential for real-world deployment.

---

> ### Author Response · Authors · 2025-11-17
> **Respond to reviwer 9USL**
>
> We appreciate the reviewer’s positive assessment of our methodology, empirical evaluation, benchmark coverage, and ablation studies. We also thank the reviewer for the constructive comments, which helped us clarify the manuscript.
>
> `Weaknesses `
> > 1. Insufficient Detail on Memory Graph Mechanics: While the memory graph is central to MIRA's design, the main text lacks operational clarity on key aspects—such as criteria for adding or pruning subgoal nodes, triggers for new LLM queries, and mechanisms for resolving conflicts between LLM suggestions and agent experience in dynamic environments. Critical implementation specifics are deferred to the appendix, and Figure 1 (the purported schematic of the graph) is absent from the main paper, impeding reproducibility and obscuring the precise novelty of the graph construction process.
>
> We thank the reviewer for pointing out the need for clearer detail in the memory-graph mechanism. We agree that the memory graph is central to MIRA, and we have added more detailed descriptions in the main text covering the criteria for adding and pruning nodes, as well as the triggers for online LLM queries. We also moved the memory-graph schematic (now Fig. 2) into the main paper and strengthened cross-references to the appendix, where the complete pseudocode and full specification of the graph with examples now appear. These revisions improve the clarity and reproducibility of the graph construction process.  Revisions and added text are highlighted in blue.
>
> ***
> > 2. Narrow Empirical Scope: All experiments are confined to low-dimensional, grid-world environments. The absence of evaluations in high-dimensional, continuous, or real-world settings (e.g., robotics, vision-based control, or multimodal tasks) limits confidence in the method’s claimed generality, despite assertions of broad applicability and memory efficiency.
>
> Please see our answer to Q2 below.
>
> ---
> ---
> `Questions `
>
> > 1. Handling Conflicting or Erroneous LLM Guidance in Memory. Could the authors clarify the exact mechanism for dynamic graph updates—specifically, how conflicting agent experience and LLM-derived subgoals/trajectory recommendations are resolved in the presence of incorrect LLM priors? What is the recourse if LLM hallucinations are initially "locked in" to memory?
>
> Thank you for the insightful question. In MIRA, the agent is not forced to follow LLM-derived trajectories. LLM guidance enters only through a utility term that softly shapes the advantage estimate, so the policy remains free to explore and the environment reward continues to drive learning. Indeed, annealing the shaping term to zero ensures that any residual inaccuracies in LLM guidance are “learned away” over training, so the final policy remains unbiased with respect to the true reward. As a result, explicit conflict-resolution rules are not required in MIRA: when LLM suggestions are inaccurate, their influence naturally diminishes through the design of the utility and memory-update mechanisms.
> If an incorrect LLM prior passes screening, it typically contributes little utility because its estimated subgoal reward or similarity is low. As training proceeds, the agent eventually encounters rollout segments with higher estimated reward for the same (sub)goal, and we update the corresponding memory entry by replacing the LLM-derived segment with the agent-derived one.
> In other situations, new agent-generated nodes may be added, causing the incorrect LLM nodes to stop being retrieved and later pruned.
> These operations ensure that agent experience naturally replaces inaccurate LLM suggestions over time.
>
> These points are explicitly clarified in the manuscript, in sections 2.1, 2.3, and 2.4. Revisions and added text are highlighted in blue.
> ***

---

> ### Author Response · Authors · 2025-11-17
>
> > 2. Scalability Beyond Grid Worlds. All experiments are conducted in discrete, grid-based environments. Have the authors attempted to apply MIRA to more complex domains—such as continuous control, vision-based robotic tasks, or high-dimensional state spaces? If not, what are the anticipated bottlenecks (e.g., graph scalability, LLM prompting overhead, or similarity computation in pixel space)? Addressing this would clarify the method's potential for real-world deployment.
>
> We did not extend MIRA to continuous-control or vision-based domains in this paper, as our focus was on establishing the framework and utility shaping in settings that can be evaluated cleanly.  Extending the method to continuous spaces and higher-dimensional domains would require additional representation-learning components and is substantial enough to warrant a separate study.
>
> In extending MIRA to higher-dimensional or continuous environments, memory scalability and similarity computation can be the main bottlenecks. Memory growth can be controlled using structured organizations, for example, storing trajectory nodes in a clustered layout and restricting retrieval to the most relevant clusters rather than the full graph.
>
> Moreover, while subgoal specification in continuous control can be nontrivial, recent LLM-guided robotics work demonstrates that semantic or high-level subgoals can be grounded through learned affordances or visual embeddings (Ahn et al., 2022; Huang et al., 2022; Singh et al., 2022). These works indicate that MIRA’s subgoal nodes can be represented as regions or targets in a learned latent space. As a result, their number need not scale with the dimensionality of the underlying state space, which helps maintain memory scalability. For similarity, the discrete matching used in grid worlds can be replaced by comparisons in a learned embedding space, using the agent’s feature extractor to produce latent state representations and computing similarity with cosine distance or a learned metric, as is standard in visual RL frameworks such as R3M (Nair et al., 2022). This provides a direct path for extending MIRA’s mechanisms to continuous or visual domains.
>
> These adaptations would allow MIRA’s core components, namely, memory, similarity scoring, and utility-shaped advantages, to extend beyond grid environments, with the underlying representations adapted to the target domain. We believe that developing and evaluating them is interesting future work, but beyond the scope of the present work. We have added a discussion of these ideas to the paper’s conclusion section. Added text is highlighted in blue.
>
> Ahn, Michael, et al. "Do as i can, not as i say: Grounding language in robotic affordances." arXiv preprint arXiv:2204.01691 (2022).
>
> Huang, Wenlong, et al. "Inner monologue: Embodied reasoning through planning with language models." arXiv preprint arXiv:2207.05608 (2022).
>
> Singh, Ishika, et al. "Progprompt: Generating situated robot task plans using large language models." arXiv preprint arXiv:2209.11302 (2022).
>
> Nair, Suraj, et al. "R3M: A universal visual representation for robot manipulation." arXiv preprint arXiv:2203.12601 (2022).
>
> ***
>
> Thank you again for the positive evaluation and the helpful comments.

---

> > ### Author Response · Authors · 2025-12-02
> >
> > To further address the reviewer’s suggestion regarding the continuous-state case, we implemented a small additional experiment comparing PPO and MIRA on Gymnasium’s MountainCarContinuous. This environment is a standard sparse-reward benchmark, as the agent receives no informative intermediate signal while attempting to reach the goal at the top of the hill. The per-step reward is dominated by a small action-penalty term $-0.1a^2$, which provides no guidance about progress toward the objective. The only meaningful positive reward is issued upon reaching the goal, making exploration and long-horizon credit assignment substantially harder than in dense-feedback control tasks. For this setting, we replace the discrete similarity function with a simple RBF kernel in the continuous state–action space to compute the utility.
> >
> > We report approximate early-stage performance for PPO and MIRA using two seeds under the same training budget (2k, 6k, and 10k iterations).
> > | Method | Return @ 2k| Return @ 6k | Return @ 10k|
> > |--------|-----------------------|-----------------------|------------------------|
> > | PPO  | –9.8 ± 15.2            | 25.7 ± 14.8            | 64.3 ± 8.3             |
> > | **MIRA** | 34.6 ± 10.7             | 57.8 ± 4.9             | 82.4 ± 2.5              |

---

### Official Review · Reviewer_tfVi · 2025-11-02

**Soundness:** 3
**Presentation:** 4
**Contribution:** 3
**Rating:** 6
**Confidence:** 3

**Summary:**

The authors propose MIRA (Memory-Integrated Reinforcement Learning Agent), a novel framework that integrates LLM guidance into RL using adaptive advantage shaping along with a structured, evolving memory graph. Empirically, MIRA outperforms standard RL and hierarchical RL baselines on a suite of sparse-reward tasks (MiniGrid, BabyAI). It achieves final performance comparable to heavily-supervised, query-intensive LLM-RL methods while requiring substantially fewer LLM queries.

**Strengths:**

[S1] The core idea of ​​“adaptive advantage shaping” is well-motivated and has a solid theoretical grouding. Compared with heavy supervision using LLM (e.g., modifying rewards, which will change the structure of the MDP and affect asymptotic convergence), advantage shaping allows for initial training under LLM guidance and gradual evolution without being limited by the imperfections of the initial LLM guidance.

**Weaknesses:**

[W1] The adaptive advantage shaping mechanism requires annealing to zero to guarantee asymptotic convergence. This makes MIRA only effective for the initial learning phase, such as accelerating exploration. It cannot sustain an advantage using imperfect LLM signals.

Besides, the annealing strategy introduces new hyperparameters ($\eta_t$ and $\xi_t$), which are crucial to performance.

**Questions:**

[Q1] How about the performance on longer runs? Will HRL gradually surpass MIRA?

---

> ### Author Response · Authors · 2025-11-17
> **Response to Reviewer tfVi**
>
> We appreciate the reviewer’s helpful comments and the positive evaluation of both the method and its presentation.
>
> `Weaknesses`
> > [W1] The adaptive advantage shaping mechanism requires annealing to zero to guarantee asymptotic convergence. This makes MIRA only effective for the initial learning phase, such as accelerating exploration. It cannot sustain an advantage using imperfect LLM signals.
>
> > Besides, the annealing strategy introduces new hyperparameters ( $\eta_t$ and $\xi_t$ ), which are crucial to performance.
>
> In an idealized setting where the LLM-derived utility were perfectly aligned with the optimal action-value function, one could in principle, keep a non-vanishing shaping weight and obtain stronger asymptotic improvement guarantees than PPO. Our setting deliberately avoids this assumption in order to account for the fact that LLMs are not always reliable (e.g., they may output incorrect reward signals). Since we cannot assume that LLMs will output accurate signals, we do not aim to use the LLM to sustain a learned policy. Instead, our goal is to use the LLM’s possibly imperfect, but likely still useful, knowledge to accelerate the initial learning phase. Mathematically, we anneal the shaping weight to ensure that the final policy remains unbiased with respect to the true reward function $\mathcal{R}$, consistent with the stability principles underlying PPO. Intuitively, such annealing ensures that any errors or suboptimalities that come from the LLM outputs are “*learned away*” over the course of the training, i.e., the final policy is unbiased with respect to the true reward. Prior analyses already emphasize that PPO’s convergence behavior depends on preserving its trusted update structure, Schulman et al. (2017) establish local monotonic improvement guarantees, while more advanced results show that global optimality emerges only under carefully controlled conditions (Liu et al., 2019; Huang et al., 2024).
> Annealing $\xi_t\to 0$ therefore acts as a principled safeguard: it preserves PPO’s long-run integrity while still exploiting the utility’s benefits during the early sparse-reward phase. Empirically, once the agent has accumulated sufficient successful trajectories, the information encoded in the memory graph is already reflected in the learned critic and policy; keeping a non-zero shaping weight at that point provides little additional benefit.
> We added a discussion of this point to the paper. Revisions are highlighted in blue.
>
>
> We agree that $\eta$ and $\xi$ are influential in our method. Fig. 14 shows that setting them to reasonable values in the [0,1] range produces stable learning; while the choice of \eta and \xi affects the speed of improvement, many settings can cluster into similar reward trajectories over the course of the training, so coarse tuning over a few values is sufficient in practice.
>
>
> Schulman, J., et al. (2017). Proximal Policy Optimization Algorithms. arXiv preprint arXiv:1707.06347.
>
> Liu, Boyi, et al. "Neural trust region/proximal policy optimization attains globally optimal policy." Advances in neural information processing systems (NeurIPS) (2019).
>
> Huang, N., et al. (2024). PPO-Clip Attains Global Optimality: Towards Deeper Understandings of Clipping. Proceedings of the AAAI Conference on Artificial Intelligence (AAAI).
>
> ***
> ***
> `Questions`
> > [Q1] How about the performance on longer runs? Will HRL gradually surpass MIRA?
>
> In our runs, the HRL baseline did not surpass MIRA, which is consistent with the behavior of Hierarchical Kickstarting (Matthews et al., 2022). The method uses fixed low-level skills together with a PPO-trained manager and includes a distillation loss that encourages the student to stay close to the teacher policy. These design choices help with early exploration, but they also constrain the asymptotic performance to that of the underlying skills and teacher. In our experiments, extended training did not lead HRL to overtake MIRA.
> We have updated Tables 3 and 4 in Appendix G with new HRL results obtained after doubling its training budget. While performance improves with additional iterations, it eventually plateaus below the return achieved by MIRA. Revisions are highlighted in blue.
>
> Matthews, Michael, et al. "Hierarchical kickstarting for skill transfer in reinforcement learning." arXiv preprint arXiv:2207.11584 (2022).
>
> ***
>
> Thank you again for the supportive assessment and the helpful comments.

---

### Official Review · Reviewer_e51T · 2025-11-03

**Soundness:** 3
**Presentation:** 2
**Contribution:** 2
**Rating:** 4
**Confidence:** 3

**Summary:**

This paper proposes to use LLM's capability of generating trajectory-based plans to guide the online training of an RL agent. Since LLMs are expensive to query, the goal is to query it only a few times while training a performant RL policy. The paper proposes to integrate LLM guidance by maintaining a graph of goals, subgoals, and trajectories, and use such a graph to compute auxiliary utility signals (similar to reward shaping) to accelerate policy learning with PPO. Experiments show that the proposed algorithm can outperform RL without LLM guidance and some other baselines.

**Strengths:**

- Using LLM to provide guidance on learning for decision making is a problem that has gained a lot of recent attention..

- The idea of using a graph to represent the current knowledge learned, including subgoals, and final goals is interesting

- Experiments in FrozenLake and MiniGrid are helpful in illustrating the utility of the proposed approach

**Weaknesses:**

- Despite having good performance in gridworld environments, the paper did not discuss the applicability of the proposed method in other RL environments, e.g. continuous control (Pendulum, Mountain Car), and stochastic transition (e.g. Pacman). I assume that in continuous control, defining subgoals can be a challenge? When one has stochastic transition, then having a trajectory-based plan may not be feasible?

Also, looking at Figure 8, it looks like significant prompt engineering is needed to allow the LLM to output useful trajectories.

- For the experiments, I see that the proposed method can improve over LLM4Teach in Distracted DoorKey, but it looks like from Table 5 that LLM4Teach slightly outperforms the proposed method. Am I missing something? Is the benefit of MIRA more on the computational side, in that it makes fewer LLM queries than LLM4Teach? A comparison between the query costs between these algorithms seems important.

- (Clarity) it would be nice if the paper can provide a full pseudocode that incorporates screening unit, the maintaining of the memory graph, the calculation of the utility signal. Some parts, e.g. the similarity function s, the \rho function, and the \hat{r}_m, and c_m in (2) are not clear to me.

- I am not sure if Theorem 1 reflects the utility of the shaped advantage. Is U_k^bonus - U_max <= 0? Then the improvement rate of the proposed algorithm can be slower than that of the PPO baseline?

- After reading the paper, I don't have a good idea about when to use MIRA(offline) versus MIRA(online). From Figure 14, it looks like MIRA (offline) does quite well despite being simpler. But from Figure 6 it looks like MIRA(online) can significantly improve MIRA(offline). Can the authors comment on this?

**Questions:**

(See questions above)

---

> ### Author Response · Authors · 2025-11-17
> **Response to Reviewer e51T**
>
> We appreciate the reviewer’s detailed feedback and the close attention given to both the main paper and the supplementary material, which have helped us substantially improve the paper.
>
> > * Despite having good performance in gridworld environments, the paper did not discuss the applicability of the proposed method in other RL environments, e.g. continuous control (Pendulum, Mountain Car), and stochastic transition (e.g. Pacman). I assume that in continuous control, defining subgoals can be a challenge? When one has stochastic transition, then having a trajectory-based plan may not be feasible?
>
> Our evaluation currently focuses on discrete gridworlds since they provide controlled settings for studying sparse-reward exploration. The core components of MIRA, namely, memory, similarity scoring, and utility-shaped advantages, are not inherently restricted to gridworlds; however, the underlying representations would need to be adapted to each environment class.
> We agree that extending MIRA to continuous-control domains introduces additional challenges and would significantly broaden MIRA’s applicability. Recent LLM-guided robotics work, however, demonstrates that subgoal abstraction in continuous spaces is feasible. Systems such as SayCan (Ahn et al. 2022), InnerMonologue (Huang et al. 2022), and ProgPrompt (Singh et al. 2022) use LLMs to propose semantic or high-level subgoals (e.g., “open the drawer,” “move to the shelf”), which are then grounded into continuous control through learned embeddings or affordance models. MIRA may be able to use these ideas to construct a memory graph for such continuous environments. While we have not evaluated this direction in the present work, the underlying mechanism is compatible with such representations. We have added a discussion of this point in the manuscript.
> For stochastic environments, MIRA does not rely on deterministic rollouts: the memory graph stores short, decision-relevant fragments rather than full trajectories. This makes the utility computation robust to nondeterministic transitions. As evidence, FrozenLake (while simpler than domains such as Pacman) in our evaluation already includes stochastic slipping dynamics, and MIRA operates reliably under this form of stochasticity (see Fig. 4).
>
> Ahn, Michael, et al. "Do as i can, not as i say: Grounding language in robotic affordances." arXiv preprint arXiv:2204.01691 (2022).
>
> Huang, Wenlong, et al. "Inner monologue: Embodied reasoning through planning with language models." arXiv preprint arXiv:2207.05608 (2022).
>
> Singh, Ishika, et al. "Progprompt: Generating situated robot task plans using large language models." arXiv preprint arXiv:2209.11302 (2022).
>
> ***
> > * Also, looking at Figure 8, it looks like significant prompt engineering is needed to allow the LLM to output useful trajectories.
>
> (now) Fig. 9 shows the full prompt for transparency, but this is not the result of extensive prompt engineering. As noted in the paper, most of the prompt content is copied from the official environment description, which includes the grid layout, action set, success conditions, and textual description provided by the benchmark itself. These are the same elements any RL baseline requires to define the task. The only task-specific, manually written additions are the output format (Python policy dictionaries) and the request for two safe trajectories. We use a single, fixed prompt per environment with no per-run or per-seed tuning. We will clarify this point to avoid the impression that significant prompt engineering is required.
> To further address the reviewer’s concern, we re-ran the FrozenLake experiment using a simpler alternative wording of the prompt. The resulting plot (Fig. 13) and prompt are included in the Appendix G, and the overall performance and reward curve remained consistent across prompt variants, demonstrating robustness to reasonable differences in prompt wording.
>
> ***

---

> > ### Author Response · Authors · 2025-11-17
> >
> > > * For the experiments, I see that the proposed method can improve over LLM4Teach in Distracted DoorKey, but it looks like from Table 5 that LLM4Teach slightly outperforms the proposed method. Am I missing something? Is the benefit of MIRA more on the computational side, in that it makes fewer LLM queries than LLM4Teach? A comparison between the query costs between these algorithms seems important.
> >
> > You are correct that in Table 5, LLM4Teach’s mean final return is sometimes slightly higher than that of MIRA; however, the difference lies within the reported error bars, so the two methods are essentially comparable in final performance. The main advantage of MIRA over LLM4Teach is indeed on the query-efficiency side rather than strictly higher asymptotic return.
> > MIRA’s memory graph allows it to rely on infrequent online queries, whereas LLM4Teach queries the LLM densely during its teaching horizon (once per state–action pair). This trade-off is reflected in Fig. 6: the right panel normalizes performance by LLM calls and shows that, under equal query budgets (per-10 and per-20), MIRA achieves higher return per query (and thus higher overall return, as the query budget is fixed) than LLM4Teach. To achieve the return depicted in the left panel of Fig. 6, without query budgets, we require approximately 30 total calls for MIRA (7 offline + 20 ± 3 online), more than 500 calls for LLM4Teach, and around 50 calls for LLM-RS. Thus, MIRA matches LLM4Teach’s performance while using substantially fewer queries.
> > To address the reviewer’s suggestion, we made the number of queries used in all three methods more explicit in the revised text. The numerical values were already reported, but the revised phrasing now surfaces them more clearly. Revisions are highlighted in blue.
> >
> > ***
> >
> > > * (Clarity) it would be nice if the paper can provide a full pseudocode that incorporates screening unit, the maintaining of the memory graph, the calculation of the utility signal. Some parts, e.g. the similarity function s, the \rho function, and the \hat{r}_m, and c_m in (2) are not clear to me.
> >
> > Thank you for the suggestion. We have revised the manuscript to substantially improve clarity around the utility computation and its components. The complete pseudocode is now included in the appendix, together with a new section that provides detailed explanations of terms in the utility (similarity \mathcal{s}, alignment \rho, etc.). We also clarified the construction and evolution of the memory graph and added pseudocode summarizing the add, update, and prune operations used to maintain it during training. In the main text, we expanded the high-level description of these components and strengthened cross-references so the flow between the main presentation and the appendix is easier to follow. Due to space limitations, the full pseudocode cannot fit in the main paper, but all essential elements are now defined. Revisions and added text are highlighted in blue.
> >
> > ***
> >
> > > * I am not sure if Theorem 1 reflects the utility of the shaped advantage. Is U_k^bonus - U_max <= 0? Then the improvement rate of the proposed algorithm can be slower than that of the PPO baseline?
> >
> > We thank the reviewer for the helpful observation. The earlier theorem provided a lower bound, which intentionally made the guarantee conservative and obscured the intended effect of the shaped advantage. In the revision, we present a surrogate-based statement aligned with the PPO objective. Because the clipping operator is monotone in the advantage and our shaped advantage satisfies \tilde A_t(s,a)\ge A_t(s,a) pointwise, the shaped surrogate is no smaller than the PPO surrogate for every policy. This yields a stronger update direction than PPO’s surrogate, \mathbb{E}[\tilde A_t]\ge\mathbb{E}[A_t] with strict inequality whenever the agent follows memory-aligned steps, which matches the faster early-stage learning as seen in Figures 4 and 5.
> > We appreciate the reviewer’s suggestion, and we believe the surrogate-based formulation now presents the intended intuition more clearly.
> >
> > ***

---

> ### Author Response · Authors · 2025-11-17
>
> > * After reading the paper, I don't have a good idea about when to use MIRA(offline) versus MIRA(online). From Figure 14, it looks like MIRA (offline) does quite well despite being simpler. But from Figure 6 it looks like MIRA(online) can significantly improve MIRA(offline). Can the authors comment on this?
>
> The two figures report different quantities, which is why they may appear inconsistent at first. (now) Fig. 16 reports wall-clock time, so MIRA(offline) takes less wall-clock time to complete 2K steps in the right panel because it makes no online LLM calls and therefore avoids latency overhead. This should not be interpreted as stronger learning: in (now) Fig. 16 (left), MIRA(offline) takes considerably longer to reach a 0.5 return (half of the total possible return in the environment) in terms of wall-clock time, despite running faster on a per-step basis, because it does not benefit from any additional online LLM information during the training and therefore needs more steps than MIRA(online). In contrast, (now) Fig. 7. measures performance in environment steps, and there MIRA(online) shows clearly higher sample-efficiency. We clarified these points in Section G.3 of the manuscript.
> Occasional online queries allow MIRA(online) to refine or correct the memory graph when the static offline guidance is insufficient, reducing failed exploration and accelerating improvement in (now) Fig. 7.
>
> When to use each variant:
> * MIRA(offline) is appropriate when run-time LLM access is restricted or latency-sensitive; it improves PPO using static memory guidance with zero query cost.
>
> * MIRA(online) is preferred when limited run-time LLM access is allowed; it improves exploration efficiency, yielding stronger learning curves (now Fig. 7), at the cost of additional wall-clock time (now Fig. 16).
>
> We added these guidelines to Section 4.2 in the manuscript.
>
> ***
> Thank you again for the helpful comments.

---

> ### Author Response · Authors · 2025-12-02
>
> To further address the reviewer’s suggestion regarding the continuous-state case, we implemented a small additional experiment comparing PPO and MIRA on Gymnasium’s MountainCarContinuous. This environment is a standard sparse-reward benchmark, as the agent receives no informative intermediate signal while attempting to reach the goal at the top of the hill. The per-step reward is dominated by a small action-penalty term $-0.1a^2$, which provides no guidance about progress toward the objective. The only meaningful positive reward is issued upon reaching the goal, making exploration and long-horizon credit assignment substantially harder than in dense-feedback control tasks. For this setting, we replace the discrete similarity function with a simple RBF kernel in the continuous state–action space to compute the utility.
>
> We report approximate early-stage performance for PPO and MIRA using two seeds under the same training budget (2k, 6k, and 10k iterations).
> | Method | Return @ 2k| Return @ 6k | Return @ 10k|
> |--------|-----------------------|-----------------------|------------------------|
> | PPO  | –9.8 ± 15.2            | 25.7 ± 14.8            | 64.3 ± 8.3             |
> | **MIRA** | 34.6 ± 10.7             | 57.8 ± 4.9             | 82.4 ± 2.5              |

---

### Author Response · Authors · 2025-11-27
**Global Response to Chairs and Reviewers**

We thank the reviewers for the time and attention dedicated to our submission. Reviewers highlighted the clarity of the contribution, the strong motivation in the problem setting, the variety of benchmarks, and the transparent ablation studies. Moreover, they found the structured memory-graph mechanism, the theoretical foundation for bounded utility shaping with vanishing influence, and the reduction in required LLM queries relative to prior LLM-enhanced RL methods, the core components of MIRA, to be well-justified and compelling.

The questions raised were constructive and helped us clarify our contributions, strengthening both the empirical evidence and the overall presentation. Across all reviews, we have addressed every comment in detail. The manuscript has been substantially revised to further improve the clarity of the proposed method and its empirical support.

Key changes to the paper (all highlighted in blue):

* **Sharpening the contributions.** We have revised the abstract, introduction, and methodology to make the role of the memory graph, the shaping mechanism, and the limited-query design clearer and easier to follow.

* **Supporting analyses.** We added sensitivity and robustness study tests (prompt robustness and screening thresholds sensitivity), extended analyses to include memory-growth measurements for different environments, and the online-only FrozenLake variant (as a matched setting to the compared baseline), as well as an updated HRL experiment as requested. We also added a continuous control environment (MountainCarContinuous) experiment for MIRA.

* **Refining technical details.** We included full formulas and pseudocode for goal alignment and confidence mapping, improved cross-referencing between the main text and the appendix, and expanded the appendix’s explanation of the utility computation.

* **Improving experimental clarity.** The experimental section now opens with explicit research questions that frame each set of results, clarifying how they address the research challenges raised in the introduction and support our contribution claims.

---

### Meta-Review · Area_Chair_m9fj · 2026-01-07

**Summary:**

**Paper Summary**

This paper studies the integration of Large Language Models (LLMs) with Reinforcement Learning (RL) to reduce query complexity during online deployment. It proposes MIRA, a framework that constructs memory graphs offline and triggers online LLM guidance only when the agent's confidence in estimating rollout utility via the graph falls below a certain threshold. The results on MiniGrid and BabyAI environments demonstrate that MIRA enables a seamless transition from LLM-dependence to autonomous decision-making, significantly enhancing learning efficiency while reducing operational costs.

---

After reading the revised paper, review comments, and author responses, the AC summarizes the paper's strengths and weaknesses below.

**Strengths**
- Technical Soundness: The methodology for leveraging LLM knowledge through a memory graph represents a unique and under-explored research direction, distinguishing it from existing works that rely on LLMs primarily for reward shaping, sub-planning, or step-by-step guidance.
- Comprehensive Ablations: The proposed framework and its specific design choices are rigorously validated through extensive experiments across diverse tasks. Most experiments include multiple runs and provide robust statistical information.
- Paper Clarity: The revised version is well-written, featuring a coherent narrative and well-depicted figures and tables that clearly communicate the core message.
- Reproducibility: The inclusion of detailed pseudo-code, implementation specifics, and environmental configurations provides strong support for researchers looking to reproduce the results.

**Weaknesses**
- Limited Evaluation: The evaluation is primarily restricted to environments with discrete action spaces; the extension of the method to continuous action spaces lacks thorough analysis. Furthermore, since only one on-policy method (PPO) was tested, there is insufficient evidence regarding its compatibility with other RL algorithms.
- Methodological Design: Concerns persist regarding the heavy reliance on complex prompt engineering and the sensitivity of the hyper-parameter annealing schedule.
- Initial Clarity Issue: The paper initially faced criticism regarding its clarity. Reviewers highlighted the need for more rigorous theoretical analysis, complete pseudo-code, and polished descriptions on the method section.

**Reviewer Concerns:**

The concerns raised by the reviewers are summarized above. Following the author response, the AC believes that many of these issues have been well-resolved and found the revised paper to be an engaging read. The AC offers the following additional suggestions and concerns for the authors to consider incorporating into their final version:

- While the pilot study on the continuous MountainCar environment is a positive step, the method would be more convincing if extensively validated on a standard continuous benchmark suite.
- The paper would be strengthened by including a discussion on different annealing strategies and identifying scenarios where the current version might fail. Furthermore, it would be beneficial to compare the proposed method with works that leverage LLMs/VLMs to directly generate reward signals [r1] or preference labels [r2] to provide more robust context.
- If applicable, please provide a scheduled plan for the release of assets, including the code and any customized environments, to ensure broader research impact.
- Due to time constraints, the AC was unable to perform a exhaustive review of the appendix (e.g., the theoretical proofs); please ensure all supplemental content is thoroughly verified for correctness in the final version.

[r1] Hong et al., VICtoR: Learning Hierarchical Vision-Instruction Correlation Rewards for Long-horizon Manipulation, ICLR 2025

[r2] Wang et al., RL-VLM-F: Reinforcement Learning from Vision Language Foundation Model Feedback, ICML 2024

**Reviewer Scores:**

The paper’s initial scores were [2, 4, 6, 6], suggesting a borderline recommendation. During the rebuttal, the authors provided detailed clarifications, additional empirical evidence, and a polished manuscript. Notably, the reviewer who initially gave a score of 2 increased it to a 4. While the remaining reviewers did not provide further feedback, the AC believes that most concerns have been effectively addressed through the updates provided in the rebuttal. It is highly probable that the reviewer who provided a 4 would have further increased their score to a 6, and that the existing 6-point evaluations would be maintained or even improved. Consequently, the AC recommends accepting this paper and encourages the authors to further refine the manuscript by incorporating the suggestions provided above.

---

### Decision · Program_Chairs · 2026-01-26

Accept (Poster)